# Super-resolution microscopy reveals functional organization of dopamine transporters into cholesterol and neuronal activity-dependent nanodomains

Troels Rahbek-Clemmensen[1,2], Matthew D. Lycas[1], Simon Erlendsson [1], Jacob Eriksen [1,2], Mia Apuschkin[1], Frederik Vilhardt[3], Trine N. Jørgensen[1], Freja H. Hansen[1] & Ulrik Gether [1,2]

Dopamine regulates reward, cognition, and locomotor functions. By mediating rapid reuptake of extracellular dopamine, the dopamine transporter is critical for spatiotemporal control of dopaminergic neurotransmission. Here, we use super-resolution imaging to show that the dopamine transporter is dynamically sequestrated into cholesterol-dependent nanodomains in the plasma membrane of presynaptic varicosities and neuronal projections of dopaminergic neurons. Stochastic optical reconstruction microscopy reveals irregular dopamine transporter nanodomains (~70 nm mean diameter) that were highly sensitive to cholesterol depletion. Live photoactivated localization microscopy shows a similar dopamine transporter membrane organization in live heterologous cells. In neurons, dual-color dSTORM shows that tyrosine hydroxylase and vesicular monoamine transporter-2 are distinctively localized adjacent to, but not overlapping with, the dopamine transporter nanodomains. The molecular organization of the dopamine transporter in nanodomains is reversibly reduced by short-term activation of NMDA-type ionotropic glutamate receptors, implicating dopamine transporter nanodomain distribution as a potential mechanism to modulate dopaminergic neurotransmission in response to excitatory input.

---

[1] Molecular Neuropharmacology and Genetics Laboratory, Department of Neuroscience Faculty of Health and Medical Sciences, University of Copenhagen, DK-2200 Copenhagen, Denmark. [2] Lundbeck Foundation Center for Biomembranes in Nanomedicine, Department of Neuroscience, Faculty of Health and Medical Sciences, University of Copenhagen, DK-2200 Copenhagen, Denmark. [3] Department of Cellular and Molecular Medicine, Faculty of Health and Medical Sciences, University of Copenhagen, DK-2200 Copenhagen, Denmark. Correspondence and requests for materials should be addressed to U.G. (email: gether@sund.ku.dk)

To obtain a detailed understanding of cellular functions and how they may be altered in disease, it is essential to achieve insight into the spatial organization of the involved molecular components and how this architecture is regulated. The dopamine transporter (DAT), for example, is a key component of dopaminergic synapses, as it mediates rapid Na$^+$-dependent reuptake of released dopamine from the terminals and thereby shapes the spatiotemporal profile of dopamine signaling[1, 2]. Consequently, tight control of both DAT distribution and activity in the dopaminergic neurons are critical for proper regulation of dopamine homeostasis and thus for governing the role of dopamine in cognition, reward and locomotor control[3]. Mechanisms underlying such regulation become even more important given that dysfunctional dopamine signaling has been linked to disorders such as Parkinson's disease, attention deficit hyperactivity-disorder, schizophrenia, bipolar disorder, and drug addiction[4]. Moreover, DAT is the primary target for psychostimulants such as cocaine and amphetamines[1, 2].

Microscopic techniques have been pivotal for our understanding of cellular mechanisms that control the distribution of individual proteins at the subcellular level. Indeed, several methods have been applied to visualize DAT, including electron microscopy (EM)[5, 6] and multiple fluorescence microscopy (FM) approaches[5, 7–9]. Application of FM has permitted overall insight into constitutive and regulated trafficking of DAT (for review see Eriksen et al.[10]), as well as it has reported abundant and generally uniform expression of DAT in dopaminergic neurons[8, 11]. EM studies similarly support that DAT is widely found in the plasma membrane of dopaminergic neurons with highest density in the plasma membrane of the axonal compartment. They also interestingly revealed that the transporter appears to be excluded from the active zones[5, 6]. Nevertheless, the important insights from EM and FM are potentially limited by low antibody labeling efficiency in EM studies, and the restricted resolution of FM. Accordingly, these approaches may have missed nanoscale heterogeneities in the subcellular distribution of the transporter that could be important for spatiotemporal control of its function. In this context it is interesting that DAT has been suggested to be sequestered into cholesterol- and glycosphingolipid-enriched plasma membrane micro domains ("membrane rafts")[12–15], and that these have been proposed to serve as "hot spots" for regulation of transporter trafficking and function including amphetamine-induced dopamine release[12, 13].

New super-resolution microscopy techniques offer a major improvement in the spatial resolution (down to 10 nm in the lateral direction), thereby enabling visualization of cellular structures at a resolution unprecedented by conventional fluorescent microscopy[16–20]. Here, we employ photoactivated localization microscopy (PALM) and stochastic optical reconstruction microscopy (STORM) super-resolution microscopy to investigate the nanoscale localization of an endogenously expressed transmembrane transporter in neurons. STORM imaging of dopaminergic neurons permitted detailed visualization of DAT distribution in neuronal projections and presynaptic varicosities. Remarkably, we find that in these structures, DAT is localized to discrete, irregular, cholesterol-dependent nanodomains. Dual-color dSTORM imaging demonstrates that the domains are adjacent to, but not overlapping with, two other key components of dopaminergic terminals, tyrosine hydroxylase (TH) and vesicular monoamine transporter 2 (VMAT2). Moreover, we provide evidence that association of DAT to cholesterol-enriched nandomains is dynamic and conceivably regulated by excitatory input, as short-term activation of NMDA-type ionotropic glutamate receptors reversibly reduces nanodomain localization of the transporter. Importantly, the DAT nanodomains are also revealed by application of STORM or PALM to Cath.a-differentiated

(CAD)[21] cells transiently expressing the transporter. Summarized, our data describe a dynamic nanodomain distribution of DAT that might enable the neuron to rapidly switch the transporter between different functional localizations and thereby optimize availability and activity of the transporter on the nanoscale in the presynaptic terminals.

## Results

**PALM reveals nanodomain distribution of DAT in CAD cells.** To study the distribution pattern of DAT by single molecule localization microscopy in live cells we fused the photoswitchable Dronpa[22] to the N-terminus of DAT and expressed the construct (Dronpa-DAT) in CAD cells. Dronpa-DAT displayed functional uptake with a $K_m$ value similar to wild-type (WT) DAT (Dronpa-DAT $0.5 \pm 0.2 \mu M$ vs. WT $0.5 \pm 0.03 \mu M$, means $\pm$ s.e.m., $n = 4$). For PALM imaging, the cells were analyzed by total internal reflection microscopy (TIRF-M) enabling imaging of the plasma membrane facing the glass surface[23]. The TIRF-M images themselves showed a strong and diffuse Dronpa-DAT signal with many scantily resolved areas and a poorly defined cell border (Fig. 1a). The reconstructed PALM images, however, revealed a much better resolved Dronpa-DAT signal with a demarcated cell border characterized by multiple filapodia-like structures (Fig. 1b). Corresponding to the plasma membrane adjacent to the glass surface, we observed that the detected localizations fro Dronpa-DAT often formed irregular clusters or "nanodomains" of various sizes as clearly seen when enlarging a region of the image (Fig. 1c, examples indicated by *red arrows*). Application of a density-based spatial clustering of applications with noise (DBSCAN) analysis, which has been used to assess clustering of other membrane proteins[33, 34], substantiated a clear nanodomain pattern for DAT with a median cluster size of ~ 70 nm in diameter (Fig. 1d). Importantly, by application of the analysis recently published by Baumgart et al.[24], we could rule out that these clusters are not the result of multiple observations of single fluorophores (Supplementary Fig. 1).

We next tested whether the nanodomain distribution seen for Dronpa-DAT was a general phenomenon for any membrane-associated protein. We analyzed Src15-Dronpa, consisting of the myristoylated N-terminus of p60$^{SRC}$ fused to Dronpa. This protein has been used as a marker of homogenous, non-raft plasma membrane[25] and, in contrast to Dronpa-DAT, the PALM images revealed a much more homogenous distribution of the detected localization with a significant lower fraction of localizations in nanodomains (Fig. 1e–h, q). To investigate the nature of the Dronpa-DAT nanodomains, we extracted cholesterol from the plasma membrane using 5 mM methyl-β-cyclodextrin (mβCD)[26] or disrupted the actin polymerization with 5 µg ml$^{-1}$ cytochalasin D (CytD). The TIRF-M revealed no apparent effect of mβCD on the Dronpa-DAT distribution; however, from the reconstructed PALM images and the DBSCAN quantification, we observed a more homogenous distribution of Dronpa-DAT and a clear reduction in the number of confined nanodomains (Fig. 1i, j, q). In contrast, CytD did not cause any apparent changes in the distribution of the Dronpa-DAT signal (Fig. 1m–q). As determined by a reversible biotinylation assay[27], the mβCD treatment had no effect on Dronpa-DAT internalization, indicating that the decreased nanodomain distribution was not caused by redistribution of the transporter away from the plasma membrane (Supplementary Fig. 2). To further confirm the role of cholesterol in the nanodomain distribution of Dronpa-DAT, we treated the Dronpa-DAT expressing CAD cells with nystatin, a cholesterol-binding polyene antibiotic, which can sequester cholesterol in the membrane and disrupt membrane rafts[28]. The PALM images showed that nystatin, like mβCD,

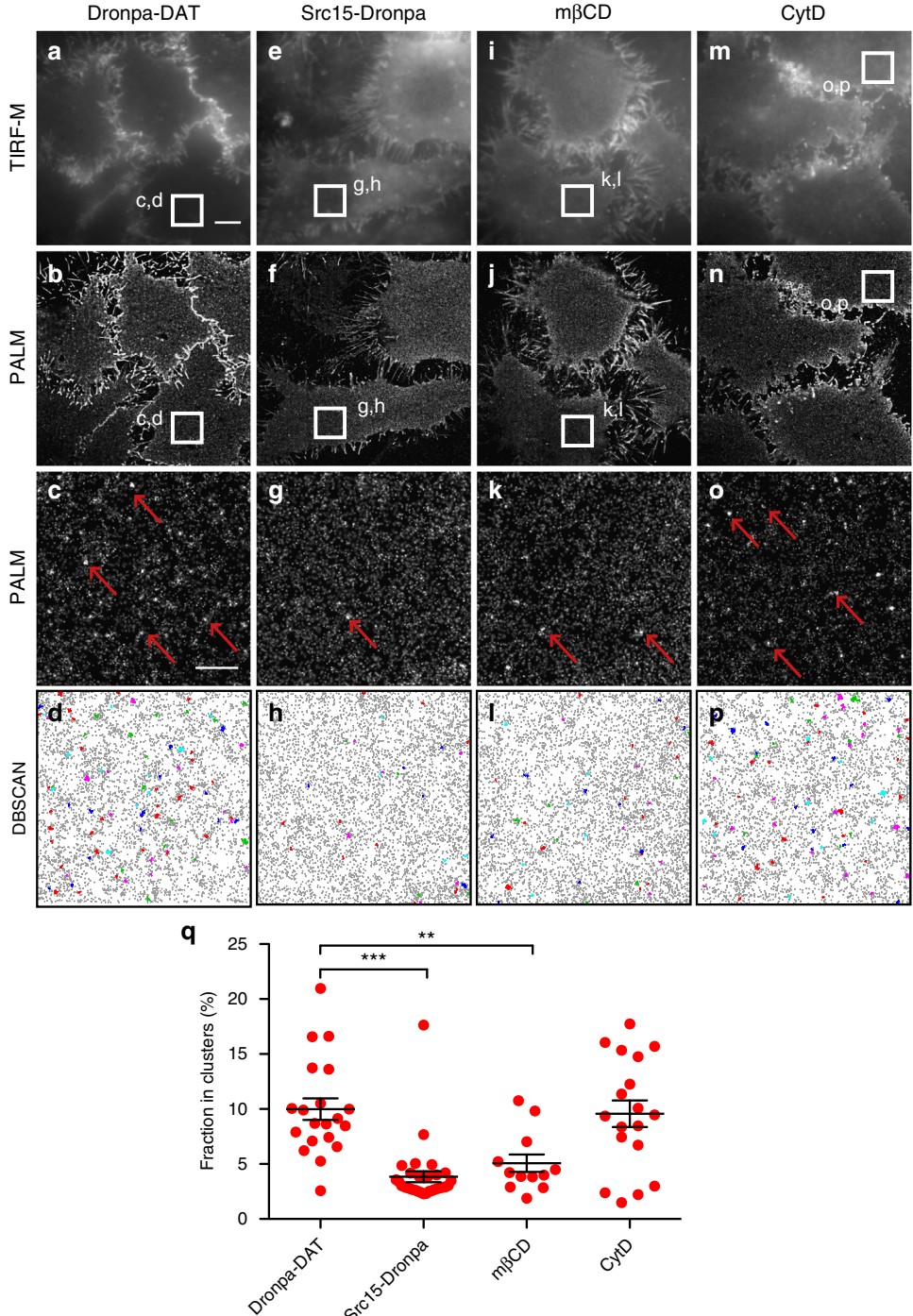

**Fig. 1** Dronpa-DAT distributes to cholesterol-dependent nanodomains in the plasma membrane of living CAD cells. **a–d** Images of untreated live CAD cells transiently expressing Dronpa-DAT. **a** TIRF-M image. **b** Reconstructed PALM image. **c** Enlarged PALM image corresponding to the boxed region in **a**, **b**. **d** DBSCAN-based cluster map of Dronpa-DAT distribution. Clustered localizations are shown by color-coding with non-clustered localizations in *gray*. **e–h** Images of live CAD cells expressing the myristoylated N-terminus of p60[SRC] fused to Dronpa (Src15-Dronpa). **e** TIRF-M image, **f** reconstructed PALM image. **g** Enlarged PALM image corresponding to the boxed region in **e**, **f**. **h** DBSCAN-based cluster map of Src15-Dronpa distribution. Clustered localizations are shown by color-coding with non-clustered localizations in *gray*. **i–l** Images of Dronpa-DAT expressing CAD cells treated with 5 mM methyl-β-cyclodextrin (mβCD) to remove cholesterol. **i** TIRF-M image, **j** reconstructed PALM image. **k** Enlarged PALM image corresponding to the boxed region in **i**, **j**. **i** DBSCAN-based cluster map of Dronpa-DAT distribution. **m–p** Images of Dronpa-DAT expressing CAD cells treated with 5 μg ml$^{-1}$ cytochalasin D (CytD) to disrupt the actin skeleton. **m** TIRF-M image, **n** reconstructed PALM image. **o** Enlarged PALM image corresponding to the boxed region in **m**, **n**. **p** DBSCAN-based cluster map of Dronpa-DAT distribution. **q** Clustering of Dronpa-DAT measured as the fraction in % of total localizations (means ± s.e.m., *$p < 0.05$, one-way ANOVA with a Bonferroni's post-test). Data are based on 12–20 cells from three independent experiments. *Red arrows* mark examples of DAT clustered in nanodomains. *Scale bars* **a**, **b**, **e**, **f**, **i**, **j**, **m**, **n** 5 μm; **c**, **d**, **g**, **h**, **k**, **l**, **o**, **p** 1 μm

caused a significant decreased in the fraction of Dronpa-DAT in nanodomains (Supplementary Fig. 3).

The marked effect of cholesterol depletion on DAT distribution prompted us to test biochemically whether DAT expressed in CAD cells is distributed into detergent-resistant membrane fractions (i.e., the membrane raft fractions) in a cholesterol-dependent manner. CAD cells transiently expressing DAT were extracted with the non-ionic detergent Brij 58 and subjected to sucrose density gradient fractionation. Western blotting analysis of the resulting fractions showed that DAT was distributed in part to the detergent-resistant high buoyancy fractions identified by flotillin-1 (FLOT-1), a marker of membrane rafts[29] (Supplementary Fig. 4a). DAT immunoreactivity was also, as expected, found in the high-density fractions identified by the transferrin receptor (TfR) (Supplementary Fig. 4a). In correspondence with our PALM analysis, cholesterol depletion with mβCD prior to cell lysis, removed DAT immunoreactivity from the detergent-resistant high buoyancy fractions, whereas CytD had no effect (Supplementary Fig. 4a). Of note, endogenous DAT solubilized

from striatal extracts also distributed in part to the detergent-resistant high buoyancy fractions (Supplementary Fig. 4b).

**STORM also shows nanodomain distribution of DAT in CAD cells.** To demonstrate that the observed nanodomain distribution of DAT in CAD cells is not a result of fusing Dronpa to the N-terminus of the transporter, we also imaged DAT in CAD cells using STORM. STORM is based on immunolabeling with fluorophore-conjugated antibodies and therefore permits visualization of a target protein without having it fused to fluorescent proteins[17]. DAT expressing CAD cells were permeabilized and immunolabeled with a rat monoclonal antibody targeting the DAT N-terminus, followed by labeling with Alexa405 and Alexa647 conjugated secondary antibodies. Equivalent to what we observed for Dronpa-DAT, the STORM images revealed localizations that often formed irregular nanodomains that were not discernible in the TIRF-M images (Fig. 2a–d). Indeed, a DBSCAN analysis of the STORM data confirmed DAT clustering in the plasma membrane of the CAD cells (Fig. 2e, k) and showed that

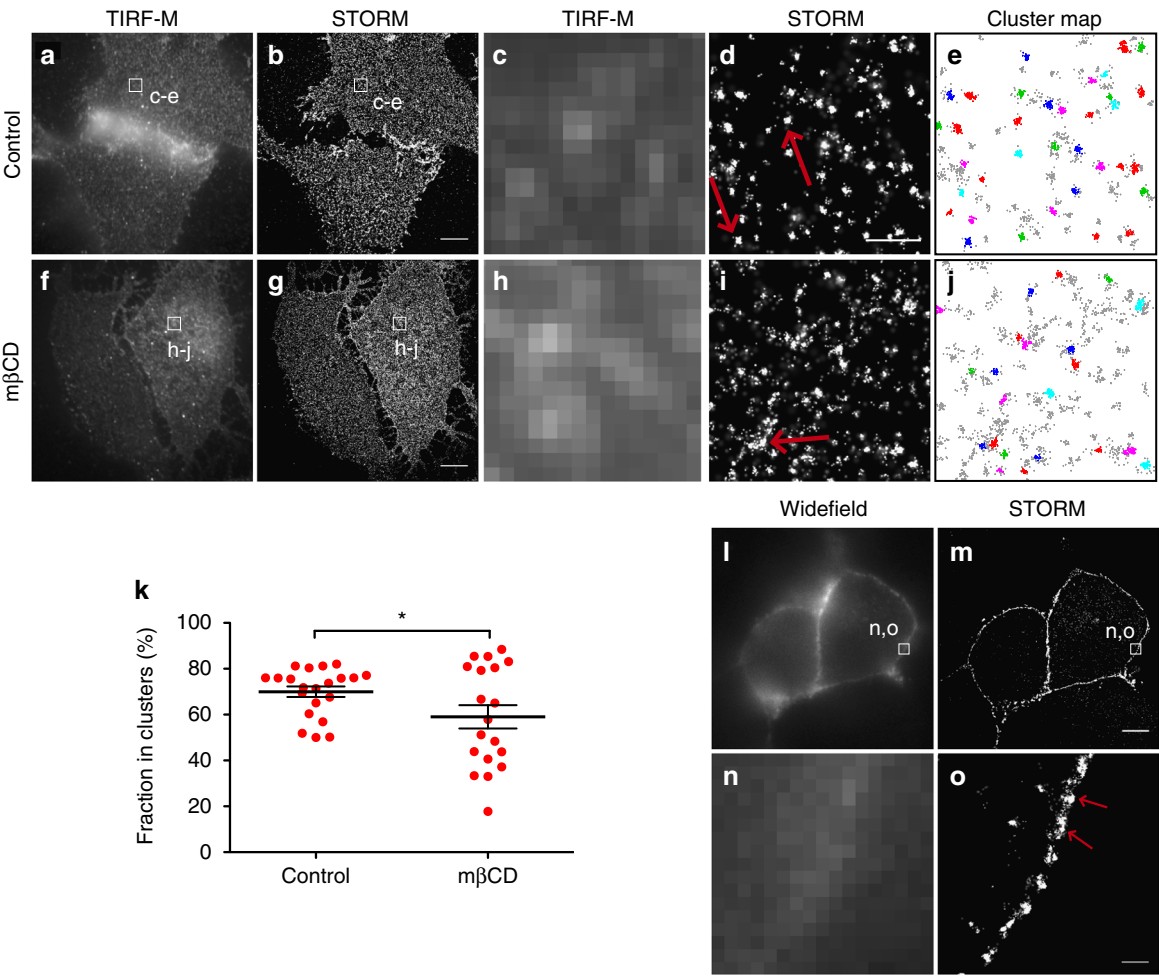

**Fig. 2** Visualization of DAT by STORM in CAD cells. TIRF-M (**a**, **c**, **f**, **h**), STORM (**b**, **d**, **g**, **i**, **m**, **o**), and widefield (**l**, **n**) images of CAD cells transiently expressing DAT with or without cholesterol depletion with 5 mM mβCD. DAT was visualized by immunolabeling with primary DAT antibody and Alexa405-Alexa647 conjugated-anti-rat secondary antibody. **a**, **f** TIRF-M view of DAT with (mβCD) or without cholesterol depletion (control). **b**, **g** Reconstructed STORM image of TIRF-M image in **a** or **f**. **c**, **d**, **h**, **i** Enlarged TIRF-M or STORM images corresponding to the boxed regions in **a**, **b** (control) or **f**, **g** (mβCD). **e**, **j** DBSCAN-based cluster map of the STORM images in **d** (control) and **j** (mβCD). Clustered localizations are shown by color-coding with non-clustered localizations in *gray*. **k** Fraction of particles (according to DBSCAN) clustered in nanodomains with (mβCD) or without cholesterol depletion (control). Data are % of localizations in cluster (means ± s.e.m., *$p < 0.05$, unpaired two-tailed *t*-test). **l–o** Visualization of DAT expressing CAD cells using inclined illumination to visualize a cross section through the cell. **l** Widefield image, **m** reconstructed STORM image of widefield image in **l**, **n**, **o** Enlarged TIRF-M or STORM images corresponding to the boxed regions in **l**, **m**. *Red arrows* mark DAT clustered in nanodomains. Data are based on 14–16 cells from three independent experiments. *Scale bars* **a**, **b**, **f**, **g**, **l**, **m** 5 μm, **c–e**, **h–j** 500 nm, **n**, **o** 400 nm

cholesterol depletion decreased the estimated fraction of DAT in clusters (Fig. 2f–k).

In addition to TIRF-M illumination, we also performed STORM on the CAD cells with inclined illumination enabling visualization of a cross section through the cell. This imaging protocol is especially important when imaging neurons that may not have larger surfaces adhering directly to the glass surface as required for useful TIRF-M data. The use of inclined illumination supported the same distribution pattern of DAT in CAD cells as that observed with TIRF-M illumination; that is, the reconstructed STORM images revealed a DAT immunosignal distributed into discrete well-defined but irregularly shaped nanodomains of various sizes along the plasma membrane (Fig. 2l–o).

**DAT is localized to nanodomains in dopaminergic neurons**. To assess the distribution of endogenously expressed DAT, we prepared cultures of dopaminergic neurons and employed a STORM protocol similar to that used for the CAD cells. By widefield microscopy, using inclined illumination to reduce background signal, we observed a diffuse DAT immunosignal in the neuronal somas with some larger clusters of intensified signal (Fig. 3a, b). The reconstructed STORM images showed a substantially better resolution with an overall uniform punctate distribution of the DAT signal, which is likely to represent DAT in various intracellular compartments, as we focus on a thin slice through the large dopaminergic soma (Fig. 3f, g). In general, we were unable to see a clear plasma membrane immunosignal in the somas, consistent with relatively low somatic plasma membrane levels of DAT and high levels of intracellular DAT[5]. Interestingly, we also observed round, often hollow, DAT-positive structures in the cytosol of the cell bodies with diameters of 100–200 nm (and thus of the same size as, e.g., clathrin-coated vesicles[30]), which could represent DAT-containing vesicles with the N-terminal tail and thus the antibody–fluorophore complex on the outside of the vesicle exposed to the cytosol (Fig. 3c, h).

In the network of neuronal extensions and varicosities along these extensions, the widefield images showed a strong DAT immunosignal that was relatively uniformly distributed with some clustered elements (Fig. 3d, e). The STORM images again revealed a more punctate DAT distribution with multiple discrete clusters along the extensions as well as in the varicosities, believed to equivalent to presynaptic transmitter release sites given the enriched presence of VMAT2[8] (Fig. 3i, j). These clusters are likely to represent DAT confined to nanodomains in the plasma membrane as the images often indicated two apparent, separated membrane sheets in the extensions (Fig. 3j). As an example, the cross-sectional line scan in Fig. 3k reveals a distance of ~ 125 nm between the two sheets shown in Fig. 3j.

A closer look at the extensions and varicosities showed that the DAT signal was clustered in irregular nanoscale domains remarkably similar to those observed by STORM and PALM in CAD cells with a median cluster size of ~70 nm in diameter (Fig. 3l, n). A DBSCAN analysis substantiated the strong DAT clustering in nanodomains along extensions and varicosities (Fig. 3m, o, p). The nanodomain distribution was further supported by a nearest-neighbor analysis[31]. If DAT is homogenously expressed, then the resulting artifact clusters would be of uniform density because the photoswitching is constant within in the image. However, the individual clusters did not have a uniform density but rather became denser at the center, strongly supporting a truly clustered distribution (Fig. 3q–t).

**DAT nanodomain localization is cholesterol-dependent**. To investigate if cholesterol also had an effect on endogenous DAT distribution, neurons were treated with mβCD before STORM

imaging. Similar to what we observed in CAD cells, mβCD decreased the clustered distribution of the DAT localizations in both neuronal extensions and varicosities (Fig. 4a–h). This was supported by DBSCAN analysis showing a significant reduction in the fraction of DAT in clusters (Fig. 4m, n). Moreover, the mβCD treatment decreased the size of the clusters, as illustrated by plotting the cumulative cluster distribution as a function of cluster area (Fig. 4o, p). In contrast to cholesterol depletion, disruption of the cytoskeleton with CytD before fixation had no effect on DAT distribution. (Fig. 4i–p). We also tested the effect of nystatin and, as in the CAD cells, we observed a decrease in DAT clustering further supporting a key role of cholesterol (Supplementary Fig. 5). The reduced clustering upon cholesterol depletion/sequestration was unlikely caused by redistribution of DAT away from the membrane. That is, we saw no change in the constitutive internalization of DAT in response to mβCD treatment, as determined by quantifying intracellular accumulation of DAT labeled with the fluorescent cocaine analog JHC 1-64[32] (Supplementary Fig. 6). Note that JHC 1-64 only visualizes surface DAT and internalized DAT, and not the large total pool of cytosolic DAT seen in the STORM images. As a consequence, the amount of DAT present on the cell surface might be less apparent in the STORM images compared to the JHC 1-64 confocal images.

**NMDA receptor activation reduces DAT nanodomain localization**. We next tested whether the nanodomain distribution of DAT was subject to dynamic regulation. Dopaminergic neurons are tightly regulated by excitatory input via ionotropic glutamate receptors. For example, a single dose of cocaine can elicit NMDA-receptor-dependent long-term potentiation (LTP) at midbrain glutamatergic synapses onto dopaminergic neurons[33, 34]. Moreover, there is evidence for the presence of NMDA receptors on the axonal terminals of these neurons, and direct short-term stimulation (5 min) of the neurons with NMDA can increase burst firing[35, 36]. We stimulated accordingly our cultured neurons either with NMDA alone (20 μM) or together with the selective NMDA receptor antagonist AP5 (100 μM), before STORM imaging. Strikingly, the resulting STORM images showed a significant decrease in DAT nanodomain localization in response to NMDA in both extensions and varicosities. This effect was blocked by AP5, in agreement with a specific effect on the NMDA receptors (Fig. 5a–p). There was no effect on the cluster size in the extensions, however, in the varicosities the NMDA treatment, similar to the mβCD treatment, decreased the cluster size of DAT (Fig. 5o, p). Subsequently, we tested whether the effect was reversible, that is, we assessed whether a 1-h recovery period following the 5-min NMDA treatment would reverse the change in DAT distribution. Again, in this experiment when neurons were fixed immediately after NMDA treatment, we reproduced the clear reduction in DAT clustering compared to the untreated neurons. However, when the dopaminergic neurons were allowed to recover for 1 h after NMDA washout, DAT recovered the degree of clustering seen in the untreated cells (Fig. 5q). Hence, NMDA appears to dynamically regulate DAT clustering in a reversible manner.

**DAT displays non-overlapping distributions with TH and VMAT2**. We finally wanted to apply dual-color dSTORM imaging to compare the distinct distribution of DAT with other key proteins of dopaminergic neurons. First, we evaluated the labeling accuracy by dual-labeling DAT with two separate fluorophores species in the same sample, Alexa647 and CF568. For this dual labeling, we observed that, although a greater number of localizations was detected with Alexa647, many of the detections from

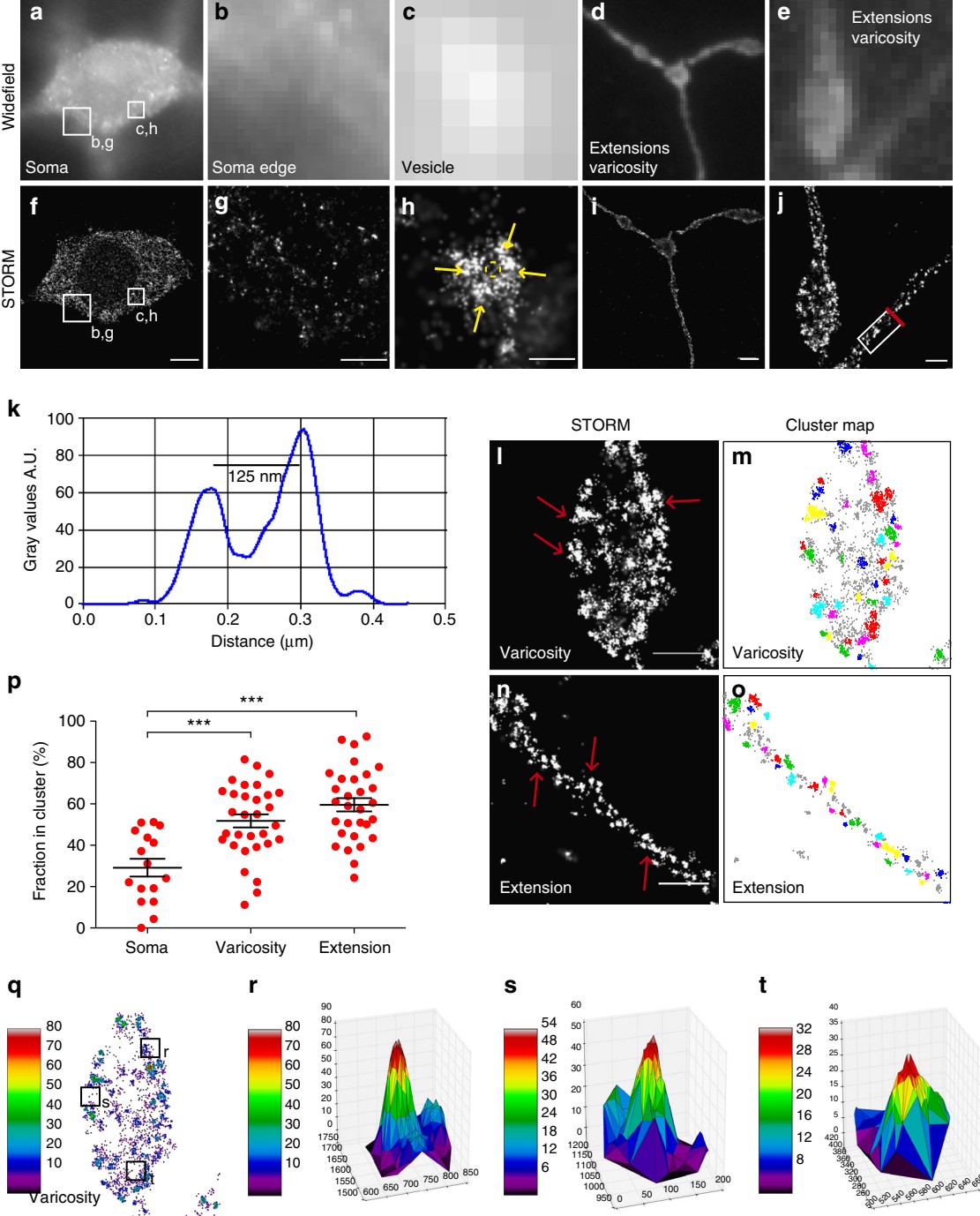

**Fig. 3** Endogenous DAT is clustered in nanodomains in the extensions and varicosities of cultured dopaminergic neurons. **a–c, f–h** Cross-section views through dopaminergic cell body showing the DAT distribution in the soma in an overall view, the soma edge, and in intracellular vesicular structures. DAT was visualized by immunolabeling with primary DAT antibody and Alexa405-Alexa647-conjugated secondary antibody. **a, f** Widefield image and corresponding reconstructed STORM image in the cell body. **b, g** Widefield image and corresponding STORM image at the soma edge; **c, h** Enlarged widefield or STORM image corresponding to the boxed regions in **a, f**, visualizing DAT in a putative endocytic vesicle. *Yellow arrows* highlight the circular distribution of the DAT signal. **d, i** Widefield image and reconstructed STORM image in a representative extension with varicosities. **e, j** Enlarged widefield image and corresponding STORM image of extension and varicosity. **k** Cross sectional profile along the boxed region in **j** showing the signal intensity as a function of the distance (μm). **l, m** STORM image and corresponding DBSCAN-based cluster map showing the degree of clustering of DAT in a representative varicosity. **n, o** STORM image and corresponding DBSCAN-based cluster map showing the degree of clustering of DAT in a representative extension. Clustered localizations are shown by color-coding with non-clustered localizations in *gray*. **p** DBSCAN-based quantification of the clustering of endogenous DAT in dopaminergic neurons in soma, varicosities and extensions. Data are % of localizations in cluster (means ± s.e.m., ***$p < 0.001$, one-way ANOVA and Bonferroni's post-test). Data are from 16 to 32 images from six individual experiments. *Red arrows* mark DAT clustered in nanodomains. **q** Varicosity from image **n**, where the localizations are colored by the number of nearest neighbors within a 30 nm radius. **r–t** Example clusters of varied densities identified in image **q**, where the number of nearest neighbors within a 30 nm radius are identified by color and by position on the z axis. *Scale bars* **a, f, l** 5 μm, **b, d, g, i** 1 μm, **c, h, m** 200 nm, **e, j, n–q, s** 500 nm

the two fluorophore species identified the same groups of DAT. This applied both to somas, where DAT appeared to be mainly in the cytosolic compartment, as also seen for the single color dSTORM images of DAT, and to extensions and varicosities, where DAT nanodomains along the presumed membrane were clearly resolvable (Supplementary Fig. 7). Importantly,

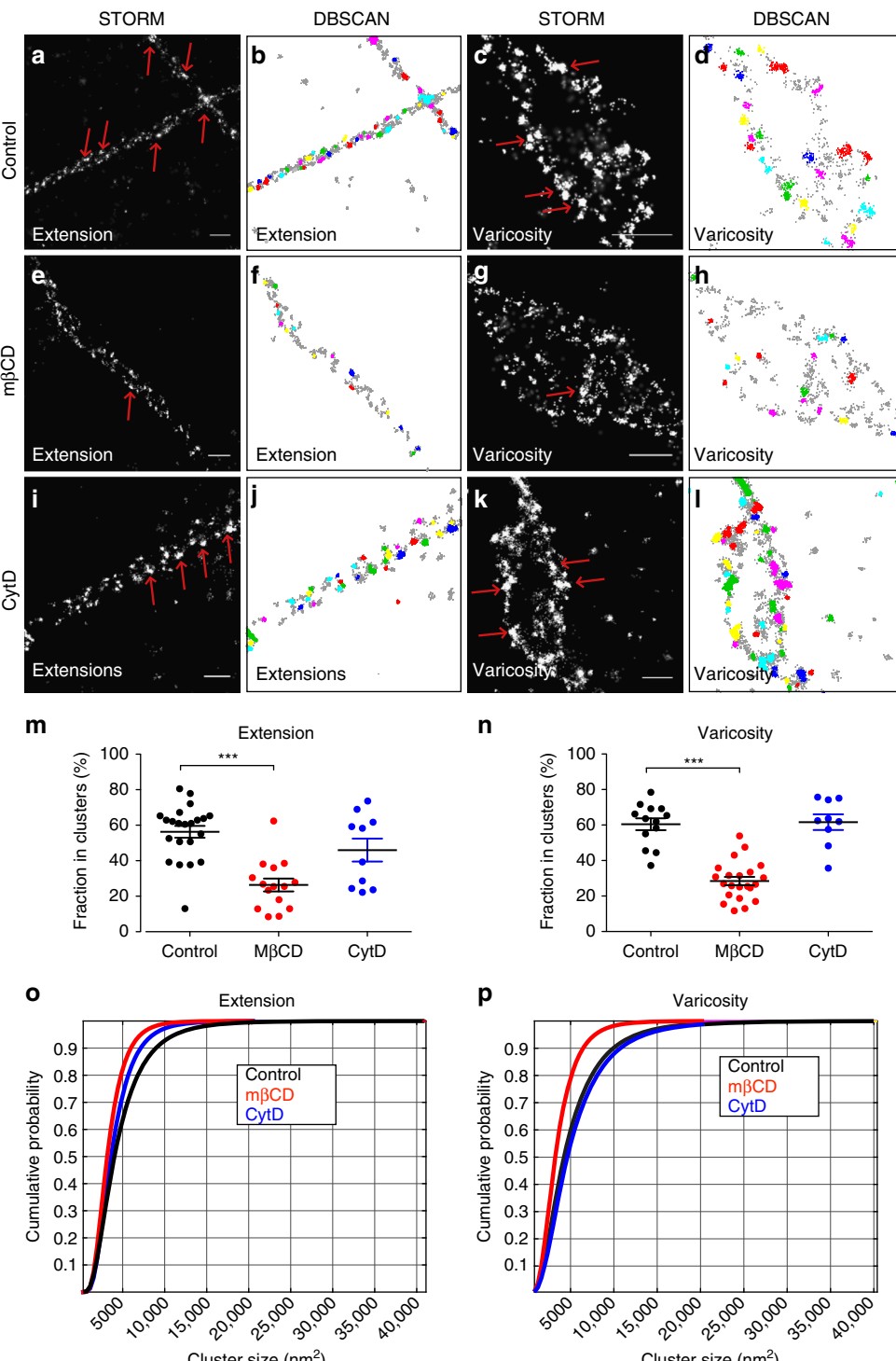

**Fig. 4** Cholesterol depletion impairs DAT nanodomain clustering in extensions and varicosities of dopaminergic neurons. **a**, **c**, **e**, **g**, **i**, **k** Representative STORM images showing DAT distribution in extensions and varicosities of dopaminergic neurons under control conditions (**a**, **c**), after cholesterol removal with 5 mM mβCD (**e**, **g**), or after actin depolymerization with 5 μg ml$^{-1}$ CytD (**i**, **k**). DAT was visualized by immunolabeling with primary DAT antibody and Alexa405-Alexa647-conjugated secondary antibody. **b**, **d**, **f**, **h**, **j**, **l** DBSCAN-based cluster maps of the DAT signal in **a**, **c**, **e**, **g**, **i**, **k**. Clustered localizations are shown by color-coding with non-clustered localizations in *gray*. **m**, **n** DBSCAN quantifications DAT nanodomain localization in dopaminergic neurons after cholesterol removal or actin depolymerization, showing **m** clustering in varicosities and **n** clustering in extensions. Data are fraction of localizations in clusters in % (means ± s.e.m., ***$p < 0.001$, one-way ANOVA and Bonferroni's post-test). **o**, **p** Probability distribution of the cluster sizes in varicosities (**o**) and extensions (**p**) at control conditions (*black*) or after treatment with mβCD (*red*) or CytD (*blue*). Data are based on from 9 to 39 images from three individual experiments. *Red arrows* mark DAT clustered in nanodomains. *Scale bars* 500 nm

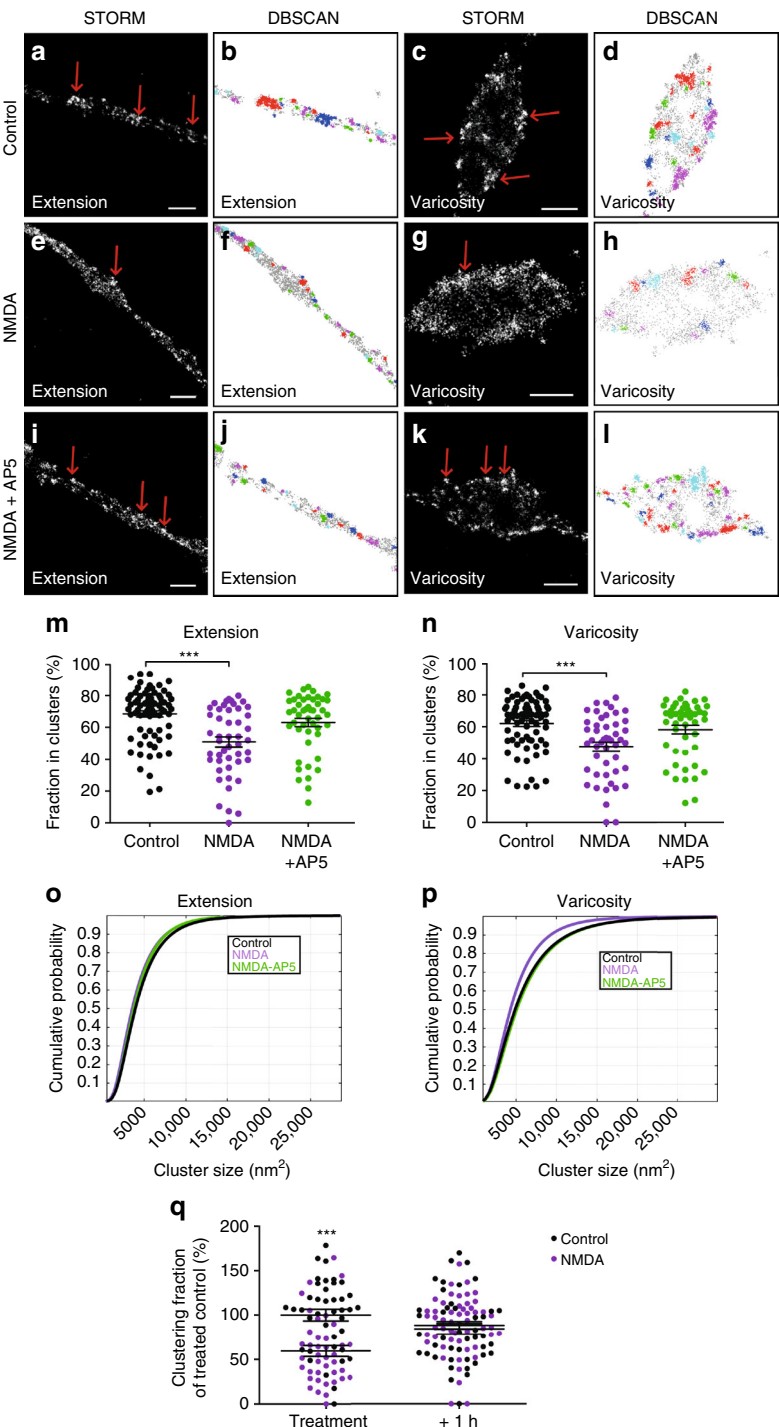

**Fig. 5** NMDA stimulation of dopaminergic neurons leads to decreased clustering of DAT. **a**, **c**, **e**, **g**, **i**, **k** Representative STORM images showing DAT distribution in extensions and varicosities of dopaminergic neurons under control conditions (**a**, **c**), after NMDA stimulation (5 min, 20 μM) (**e**, **g**), or after NMDA stimulation (5 min, 20 μM) in presence of the NMDA receptor antagonist AP5 (100 μM) (**i**, **k**). DAT was visualized by immunolabeling with primary DAT antibody and Alexa405-Alexa647-conjugated secondary antibody. **b**, **d**, **f**, **h**, **j**, **l** DBSCAN-based cluster maps of the DAT signal in **a**, **c**, **e**, **g**, **i**, **k**. Clustered localizations are shown by color-coding with non-clustered localizations in *gray*. **m**, **n** Quantifications, based on the DBSCAN, of DAT nanodomain localization in dopaminergic neurons after NMDA or after NMDA + AP5 showing **m** clustering in varicosities, and **n** clustering in extensions. Data are fraction of localizations in clusters in % (means ± s.e.m., ***$p < 0.001$, one-way ANOVA and Bonferroni's post-test). **o**, **p** Probability distribution of the cluster sizes in varicosities (**o**) and extensions (**p**) at control conditions (*black*) or after NMDA (*purple*) or after NMDA + AP5 (*green*). Data are from 63 to 64 images from four individual experiments. *Scale bar* 200 nm. **q** Reversibility of NMDA-induced decrease in DAT nanodomain localization. Neurons were treated for 5 min with NMDA (*purple*) or vehicle (*black*) and fixed (treatment) or treated for 5 min with NMDA (*purple*) or vehicle (*black*) followed by a 1 h wash out before fixation (+1 h). Data are based on DBSCAN of the resulting STORM images and shown as fraction of localizations in clusters after NMDA as percentage of fraction of localizations in clusters at control conditions (means ± s.e.m., ***$p < 0.001$, two-way ANOVA and Bonferroni's post-test). Data are from 146 images from two individual experiments. *Scale bars* 500 nm

a quantitative co-localization analysis further supported the overall strong overlap in labeling between the two fluorophores (Supplementary Fig. 7).

Tyrosine hydroxylase is the rate-limiting enzyme in dopamine synthesis and an abundant protein in dopaminergic neurons. Dual-color dSTORM, targeting DAT and TH with Alexa647 and CF568, respectively, clearly revealed distinct patterns of distribution for the two proteins. In the cytosol of cell bodies, the DAT and TH signals were separated with very modest, if any, overlap (Fig. 6a–f). In the extensions, a punctate appearance of the TH signal was found between the membrane sheets demarcated by the DAT signal (Fig. 6g–l). Corresponding to the varicosities, we observed a strong TH signal unequally distributed in the presumed cytosol, sometimes forming larger confluent areas while at the same excluding other areas. Again, the TH signal was not overlapping with DAT, but, was often seen immediately adjacent to the conceivably plasma membrane located DAT (Fig. 6m–o).

The vesicular monoamine transporter (VMAT2) sequesters monoamines, including dopamine, into synaptic vesicles[4]. Dual-color dSTORM targeting DAT and VMAT2 showed that in the somatic cytosol both proteins were relatively uniformly distributed, although the VMAT2 localizations appeared in larger groups compared to DAT. Neither DAT nor VMAT2 appeared to strongly accumulate in the somatic plasma membrane and, strikingly, we saw essentially no overlap between the two proteins, supporting differential cellular sorting of the two membrane proteins (Fig. 7a–f). In the neuronal processes, the VMAT2 signal was markedly enriched in the varicosities (and thus at the presumed presynaptic release sites) with some localization to the extensions where a punctate signal was present between the two apparent membrane sheets defined by the DAT distribution (Fig. 7g–i). Corresponding to the varicosities, the strong VMAT2 signal often appeared almost confluent in the cytosol, though in other cases a clear polarization of signal was observed. The VMAT2 signal in varicosities was sometimes found to reside in immediate vicinity of a large population of DAT, but there was very little overlap in the signals (Fig. 7g–o). Altogether, these dual-color experiments substantiate the differential distribution pattern at sub-diffracting resolution of three key components of the dopaminergic presynapse.

## Discussion

Visualization of cellular processes is of fundamental importance to gain insight into complex biological systems. Indeed, visualization with classical FM has proven its value when studying many cellular processes, such as those governing targeting and trafficking of receptors, transporters and ion channels in different cell types. Nonetheless, the diffraction-limited resolution obtainable with conventional FM has prevented visualization of small cellular structures with sizes in the lower nanometer range, and thereby hindered a detailed dissection of the molecular distribution of membrane protein on the nanoscale and how this is regulated. With the advent of super-resolution microscopy, permitting fluorescent imaging at subdiffraction resolution, a substantial part of these obstacles has been overcome, as illustrated by the application of this technique to DAT in the present study. Although EM still offers superior resolution, technical difficulties and requirements for sample preparation challenge the applicability of EM when testing the effect of pharmacological and biochemical manipulations. Specifically, the low labeling efficiency of immunogold EM imaging makes the technique less suitable for investigating protein clustering in a quantitative manner[37].

In this study, we apply super-resolution microscopy to reveal and visualize a dynamic molecular distribution of DAT in dopaminergic neurons. Specifically, the super-resolution images revealed that DAT is not uniformly distributed in the plasma membrane, as seen with classical microscopy[8, 11, 38], but localized to multiple irregular clusters or "nanodomains". The nanodomains were clearly seen in the neuronal extensions where they demarcated two separate membrane sheets only 100–150 nm apart, as well as they were strongly present in the varicosities that presumably represent presynaptic transmitter release sites[8]. Corresponding to the cell bodies of the cultured neurons, we observed little apparent DAT signal in the plasma membrane, but a strong cytoplasmic DAT signal that may represent newly synthesized DAT in the ER and Golgi in agreement with previous findings[5].

The DAT nanodomains in the extensions and varicosities of the dopaminergic neurons were also seen upon expression of DAT in catecholaminergic CAD cells. In the CAD cells, the DAT nanodomains were visualized both by STORM, as in the neurons, and by PALM. Whereas STORM and dSTORM exploit secondary antibody-amplification together with a photoswitchable multi-emitter fluorophore, PALM is based on use of photoswitchable fluorescent protein fused to the protein of interest, in this case, Dronpa fused to DAT. PALM can accordingly be performed on live cells without fixation and permeabilization. Moreover, the 1:1 stochiometry between the fluorescent protein and the target protein abolishes putative artifacts derived from having more than one or less than one fluorophore attached to the target protein[17]. The need for using a fusion protein in PALM prevents analysis of endogenously expressed proteins[17]. STORM/dSTORM, on the other hand, excels by allowing investigation of endogenously expressed proteins, but typically requires fixation and, if the epitope is intracellular, permeabilization[37]. Importantly, we observed nanodomain distribution of DAT with STORM in the CAD cells as we did with live PALM, suggesting that fixation and permeabilization are not a problem for our observations. We should note, however, that that the apparent fraction of DAT in clusters was higher according to our STORM data than in our PALM data. A conceivable explanation for this apparent discrepancy might relate to differences in the amount of blinking per fluorophore and the stoichiometry of the fluorophores to transporter, which in the DBSCAN analysis may lead to a higher estimated clustered fraction in STORM compared to PALM.

The confinement of DAT into nanodomains displayed a striking dependency on cholesterol. Both removal of cholesterol with mβCD or sequestering cholesterol with nystatin markedly reduced DAT nanodomain localization in neurons and CAD cells. This observation is not least interesting because the high-resolution structure of Drosophila DAT revealed a cholesterol-binding site on the surface of the transporter[39]. In addition, cholesterol has been shown to regulate DAT function[14, 15, 40] as well as the lateral mobility of heterologously expressed DAT in the plasma membrane[13, 14]. It is thus tempting to suggest that cholesterol-dependent sequestration of DAT into nanodomains represents a spatial correlate of function governed by direct binding of cholesterol to the transporter. The influence of cholesterol on DAT nanodomains is moreover of interest in light of previous biochemical data, as well as biochemical data from this study, indicating that DAT, in heterologous cells and striatal tissue, segregates into detergent-resistant membrane fractions (membrane raft factions) dependent on cholesterol[12, 14, 15]. The nanodomain distribution of DAT could also have important implications in relation to findings suggesting that localization of DAT to membrane cholesterol-dependent rafts is important for amphetamine-induced dopamine efflux via DAT[12, 41] and for regulation of transporter internalization[15]. Even more intriguing and with potentially larger functional implications, we found that

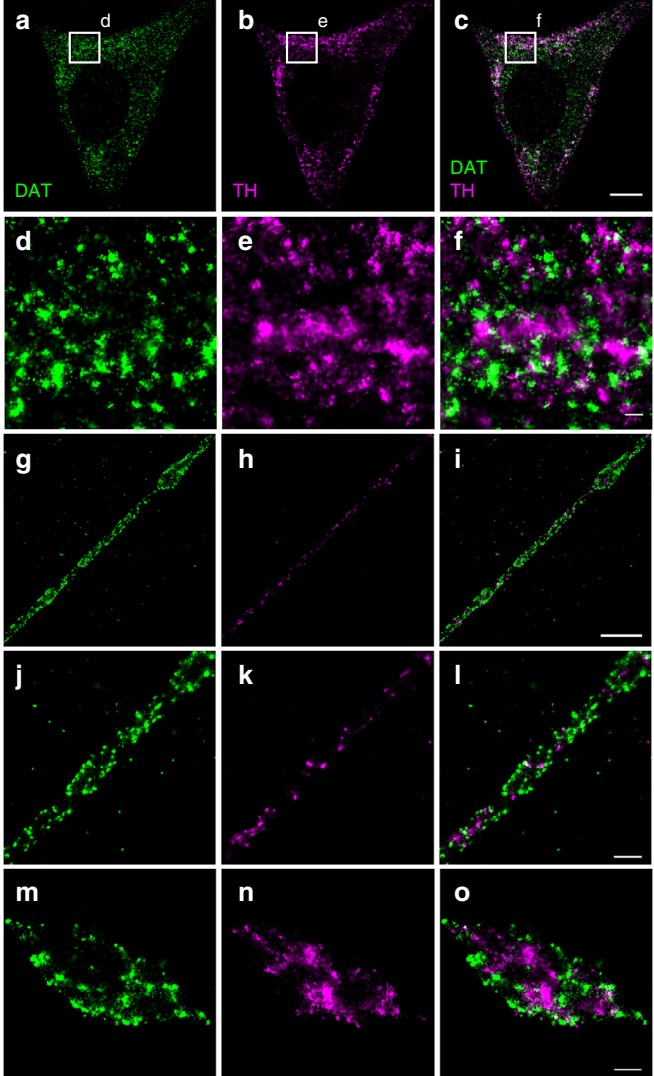

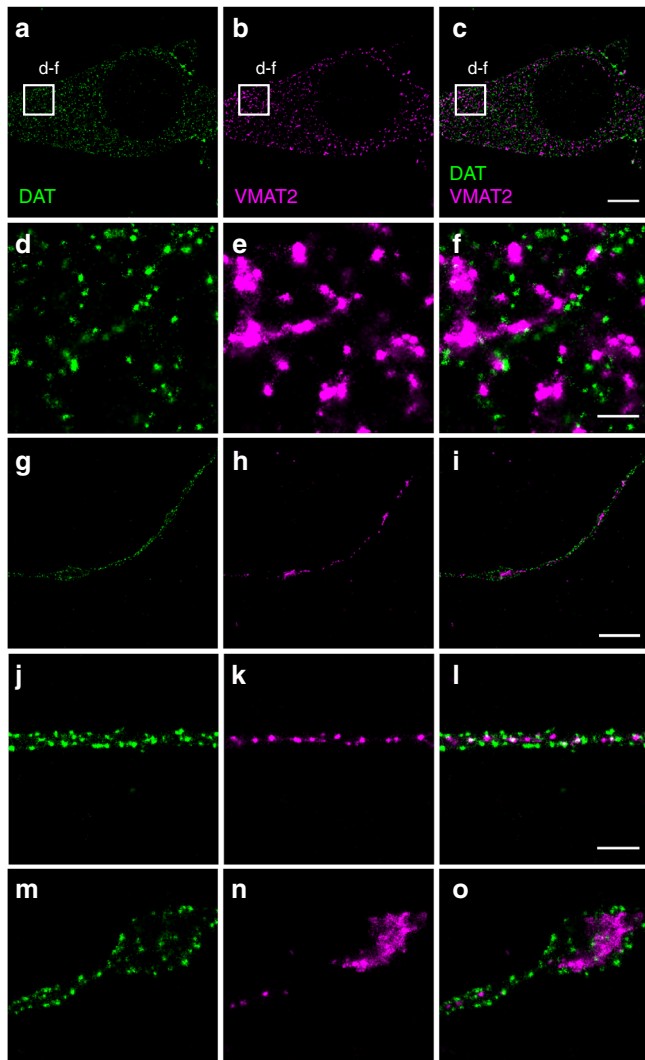

**Fig. 6** Cross-section through the cell body, extensions and varicosities of dopaminergic neurons showing by dual-color dSTORM imaging the distinct distribution of tyrosine hydroxylase (TH) compared to DAT. DAT was visualized by immunolabeling and Alexa647-conjugated secondary antibody. TH was visualized by immunolabeling and CF568-conjugated secondary antibody. **a** Somatic distribution of DAT. **b** Somatic distribution of TH. **c** Merged image of **a** and **b**. **d–f** Enlarged STORM images corresponding to the boxes shown in **a–c**. **g** Distribution of DAT in extension with varicosity. **h** Distribution of TH in extension with varicosity. **i** Merged image of **g** and **h**. **j** Distribution of DAT in extension. **k** Distribution of TH in extension. **l** Merged image of **j** and **k**. **m** Distribution of DAT in varicosity. **n** Distribution of TH in varicosity. **o** Merge image of **m** and **n**. *Scale bars*: **a–c** 5 μm, **g–i** 2 μm, **d–f**, **j–o** 500 nm

**Fig. 7** Cross-section through the cell body, extensions and varicosities of dopaminergic neurons showing by dual-color dSTORM imaging the distinct distribution of the vesicular monoamine transporter-2 (VMAT2) compared to DAT. DAT was visualized by immunolabeling and Alexa647-conjugated secondary antibody. VMAT2 was visualized by immunolabeling and CF568-conjugated secondary antibody. **a** Somatic distribution of DAT. **b** Somatic distribution of VMAT2. **c** Merged image of **a** and **b**. **d–f** Enlarged STORM images corresponding to the boxes shown in **a–c**. **g** Distribution of DAT in extension with varicosities. **h** Distribution of VMAT2 in extension with varicosities. **i** Merged image of **g** and **h**. **j** Distribution of DAT in extension. **k** Distribution of VMAT2 in extension. **l** Merged image of **j** and **k**. **m** Distribution of DAT in varicosity. **n** Distribution of VMAT2 in varicosity. **o** Merge image of **m** and **n**. *Scale bars* **a–c** 5 μm, **g–i** 2 μm, **d–f**, **j–o** 500 nm

the cholesterol-dependent nanodomain distribution of DAT is sensitive in a reversible manner to activation of excitatory receptors, such as NMDA receptors. NMDA receptor activation on dopaminergic neurons is known to increase burst firing[43, 44] and, on this background, it is tempting to suggest that reversible association to cholesterol-dependent nanodomains might represent a means by which the neuron rapidly can adjust transporter localization and availability on the nanoscale in order to adapt to neuronal activity.

Previous biochemical data have suggested a tight association between DAT, TH and VMAT2 containing synaptic vesicles[42, 43]. For example, it has been proposed that DAT and VMAT2 are

physically linked together via the synaptic vesicle protein synaptogyrin-3[43]. To our knowledge, the relative distribution of these proteins, however, has never been investigated at sub-diffracting resolution. Importantly, the present dual-color super-resolution experiments demonstrate a distribution of DAT that is clearly distinct from that of both VMAT2 and TH. TH, the rate-limiting enzyme in dopamine synthesis, showed an irregular distribution in the cytosol of both extensions and varicosities. In the varicosities, the clustered signal was often found immediately adjacent to, but not overlapping with, the DAT nanodomains. This supports close proximity, but not formation of larger co-complexes. For VMAT2, we found a strongly enriched signal in

the varicosities with limited signal in the extensions, consistent with the varicosities representing presynaptic release sites. We also observed that the VMAT2 signal was more uniform than the TH signal in the varicosities and in several cases "polarized", i.e., the signal was localized essentially to only one part of the varicosity. Nonetheless, similar to what we saw for TH, the VMAT2 signal was often immediately adjacent to, but again, not overlapping with the DAT signal, arguing against formation of larger co-complexes.

Still relatively few studies have exploited super-resolution microscopy to visualize neuronal membrane proteins[31, 44–53] and, to our knowledge, no other study has reported sequestration of an endogenously expressed neuronal membrane protein into cholesterol-sensitive nanodomains that might be sensitive to excitatory input via activation of, e.g., NMDA-receptors. Application of STORM has allowed a description of how e.g. ionotropic glutamate receptors themselves as well as metabotropic GABA receptors are positioned in the pre- and post-synapse[44–48]. Super-resolution techniques have also suggested compartmentalized distribution of several membrane proteins into nanodomains[31, 50, 54–57] as well as formation of super-complexes of ion channels[52]. AMPA receptors were found to cluster in spines[46] and to be enriched in dynamic nanoscale scaffolding domains of PSD-95[57]. It has also suggested that the SNARE protein syntaxin 1 is sequestered in distinct nanoscale clusters in the *Drosophila* neuro-muscular junctions and in heterologous PC12 cells but it was not assessed whether the clusters were dependent on cholesterol and/or the cytoskeleton[31, 54]. Yet other neuronal membrane proteins may not cluster such as the cannabinoid CB1 receptor, adopting a uniform presynaptic distribution in GABAergic interneurons[50].

In summary, the present study demonstrates the power of super-resolution microscopy to dissect the distribution of an important membrane transport protein in neurons. Specifically, we were able to visualize in detail how DAT is dynamically sequestered into irregular nanodomains in cellular structures, such as presynaptic varicosities, which cannot be efficiently resolved by classical FM due their small size. By revealing a striking cholesterol dependency and sensitivity to excitatory input, our results introduce new means by which DAT function can be regulated by membrane lipids and neuronal activity on the nanoscale. Furthermore, our data represent an important framework for future studies aimed at further dissecting the dynamic molecular architecture of dopaminergic neurotransmission and how this might change in diseases characterized by dysfunction of the dopamine system.

## Methods

**DNA constructs**. Dronpa was a kind gift from Dr. Atsushi Miyawaki Brain Science Institute, RIKEN, Japan[58]. EGFP was cut out or the pEGFP-C1 vector (Clontech), using *AgeI* and *BsrGI*, and replaced with a PCR fragment encoding Dronpa to generate pDronpa-C1 (sense primer:TCCGCTAGCGCTACCGGTCGCCACCA TGAGTGTGATTAAACCAG; antisense primer:ACTTGTACAGCTTGGCCTG CCTCGGCAGC). Subsequently, hDAT was subcloned from pcDNA3.1 into the KpnI and XbaI sites of the pDronpa-C1 generating pDronpa-DAT. The construct was verified by sequencing. Src15-Dronpa, consisting of the myristoylated N-terminus of p60[SRC] fused to Dronpa was synthesized by DNA 2.0 (Menlo Park, CA). For sucrose gradient fractionation experiments, we used hDAT encoded by a synthetic gene kindly provided by Dr. Jonathan Javitch, Columbia University, NY, USA[59].

**Cell culture**. CAD (purchaseable from Sigma, 08100805) cells were grown in media containing Ham's F12 and DMEM (1:1) (GIBCO, Grand Island, NY) and 10% (w/v) fetal bovine serum (FBS) (Invitrogen, Carlsbad, CA) and 1% (v/v) penicillin/streptomycin (v/v), in 5% CO$_2$ and 95% humidified atmosphere at 37 °C. Cells were transiently transfected 48 h prior to experiments in a 1 μg:3 μl Lipofectamine 2000 (Invitrogen) ratio according to manufacturer's protocol. The cells were regularly tested for mycoplasma with a standard mycoplasma test kit (Lonza, NJ, USA).

**Dopamine uptake**. One day prior to the experiment, 100,000 transiently transfected CAD cells were seeded in 24 well plates (TPP, Trasdingen Switzerland). On the day of the experiment, the cells were preincubated for 10 min in a HEPES buffered saline (HBS) (25 mM HEPES, 120 NaCl, 5 mM KCl, 1.2 mM CaCl$_2$, 1.2 mM MgSO$_4$, 1 mM ascorbic acid, 5 mM D-glucose and 1 μM Ro 41-0960 (COMT inhibitor) pH 7.4. Subsequently the cells were incubated for 3 min at room temperature with a serial dilution from 6.4 to 0.05 μM $^3$H-DA (2,5,6,7,8-[$^3$H]-DA 52.8–61.4 Ci mmol$^{-1}$, PerkinElmer Life and Analytical Sciences, Boston, MA). Uptake was measured using Optiphase HiSafe 3 scintillation fluid (PerkinELmer Life and Analytical Sciences).

**Dissection and culturing of dopaminergic neurons**. Dopaminergic neurons from the ventral midbrain were isolated using a protocol modified from Rayport et al.[60]. The ventral tegmental area and substantia nigra were located and isolated from P1 to P2 Wistar rats (Charles River, Germany) and dissolved in a papain solution (116 mM NaCl, 5.4 mM KCl, 26 mM NaHCO$_3$, 2 mM NAH$_2$PO$_4$, 1 mM MgSO$_4$, EDTA, 25 mM glucose 1 mM cysteine, 0.5 mM Kyrunate, and Papain 20 U ml$^{-1}$) for 25 min at 37 °C. The tissue was gently triturated to a single cell suspension by mechanical force and spun for 10 min at 100 × g, and seeded in warm SF1C [50% (v/v) modified Eagle's medium (MEM), 40% (v/v) Dulbecco's modified eagles medium (DMEM), 10% (v/v) F-12 (all from Invitrogen) supplemented with 1% (v/v) heat inactivated calf serum (FBS, 2.5 mg ml$^{-1}$, Invitrogen), 0.35% (w/v) D-glucose, 0.5 mM glutamine, 5 mM Kyrunic acid, Penicillin, Streptomycin, liquid catalse (0.05%), and DiPorzio[61] (containing insulin, transferrin, superoxide dismutase, progesterone, cortisol, Na$_2$SeO$_3$, and T3)] on Poly-D-Lysine coated, 1 M KOH sonicated coverslips with a thickness of 160–180 μm for STORM imaging or on a glia cell monolayer on coverslips for internalization experiments. Two hours after seeding, the cells were treated with 10 ng ml$^{-1}$ glia derived neurotrophic factor (GDNF). 5-Fluorodeoxyurdine was added to the cell media at 6–7 DIV in order to inhibit cell proliferation of glia cells. The sample was used after 8–11 DIV.

**Antibody labeling**. Antibodies were labeled using a protocol described in Bates et al.[62]. The antibody (63 μg) was incubated for 40 min while shaking at room temperature with 100 mM NaHCO$_3$, Alexa647 (17 μg ml$^{-1}$), Alexa405 (50 μg ml$^{-1}$) (Invitrogen, Carlsbad, Germany), or CF568 (Biotium, Freemont, CA) in a total volume of 60 μl in PBS and afterwards mixed with 140 μl PBS on a NAP-5 column (GE Healthcare) to isolate conjugated antibody–fluorophore complexes. All fluorophore-conjugated antibodies used were tested in order to keep the relative levels of fluorophore:antibody steady. For single color DAT labeling the Alexa405: Alexa647:antibody ratio was [2,3]:[0.8–0.9]:1. For dual color dSTORM the Alexa647:antibody or CF568:antibody ratio was [0.8–0.9]:1.

**Immunocytochemistry**. To assess the role of cholesterol and the cytoskeleton, dopaminergic neurons or transfected CAD cells were treated with 5 mM methyl-β-cyclodextrine (mβCD) for 30 min, 10 μg ml$^{-1}$ nystatin for 20 min or 5 μg ml$^{-1}$ cytochalasine D (CytD) for 2 h in cell media at 37 °C. To investigate the effect of NMDA stimulation, the neurons were incubated for 9 min in imaging buffer with or without 100 μM of the NMDA receptor antagonist AP5 before treatment for 5 min at room temperature with imaging buffer (control), 20 μM NMDA in imaging buffer, or 100 μM AP5 and 20 μM NMDA in imaging buffer. Cells were fixed using 3% (w/v) paraformaldehyde (PFA) in PBS (Electron Microscopy Sciences, Pennsylvania, USA) and incubated in 50 mM NH$_4$Cl (Sigma) to quench background fluorescence. The sample was blocked and permeabilized in 5% (w/v) goat serum (Gibco), 0.2% (w/v) Saponin (Sigma) in PBS, and subsequently incubated for 1 h in in the appropriate primary antibody, DAT: rat anti-DAT (Millipore, mab369, Massachusetts, USA) 1:200; TH: rabbit anti-TH antibody (Invitrogen, OOPA1-04050) (1:1000), VMAT2: rabbit anti-VMAT2 antibody (1:2000) (a kind gift from Dr. Gary Miller, Emory University)[63] followed by secondary antibody 5 μg ml$^{-1}$: Donkey anti-Rat (Jackson ImmunoResearch, 712-005-150, Pennsylvania, USA) or Donkey anti-Rabbit (Jackson ImmunoResearch, 711-005-152) conjugated with indicated fluorophores. The sample was post-fixed in 3% PFA in PBS and stored cold in PBS.

**Striatal synaptosomal preparations**. Coronal sections (1.5 mm) were used to isolate the striatum from 9 to 11 weeks old mice. The tissue was homogenized in homogenization buffer (0.32 M sucrose, 4 mM HEPES, pH 7.4) using a motor driven Teflon pestle at 800 r.p.m. The membranes were isolated by a dual spin; 1000 × g for 10 min at 4 °C to remove cell nuclei and 16,000 × g for 20 min at 4 °C to pellet the membranes and remove cytoplasma.

**Sucrose gradient**. CAD cells transiently expressing DAT, treated as explained above, or striatal synaptosome preparations from 9 to 11 weeks old mice were lysed in 1 ml lysis buffer (1% (w/v) Brij58 (w/v), protease inhibitor, and 0.2 mM PMSF in the gradient buffer [25 mM HEPES, 150 mM NaCl, pH 7.4]) for 15 min on ice. A 14 ml continuous sucrose size gradient was made by mixing 1 ml sample in a 15 ml centrifuge tube (Beckman Coulter, California, USA) with 1 ml sucrose (80% w/v) in gradient buffer to create 2 ml with final 40% (w/v) sucrose content. On top of this a 12 ml 35–15% (w/v) continuous gradient was layered using a SG15 gradient maker (Hoefer, Massachusetts, USA). Subsequently the gradient was ultracentrifuged at

100,000 × g for 18 h in a Beckman SW28.1 rotor head (Beckman Coulter, USA) and fractionated into nine sections of 1.5 ml using a P1 pump. The samples were electrophoresed using a 15 well, any kD, pre-cast gel (Biorad, California, USA), transferred on to a PDVF membrane (Millipore, Massachusetts, USA) and developed with various primary antibodies; rat anti-DAT (Millipore, mab369) 1:1000, rabbit anti-Flotillin1 (Sigma, Missouri, USA) 1:1000, mouse anti-transferrin receptor (Invitrogen, 13-6800), and mouse-NaK ATPase (Abcam, ab7671, United Kingdom) 1:250 in 5% (w/v) milk powder, 0.05% (w/v) Tween-20 in PBS. Secondary antibodies were HRP-conjugated goat anti-rat (Pierce, 31,470, Illinois, USA), and donkey anti-rabbit (Pierce, 31,458) and goat anti-mouse (Thermo Scientific, 31,430) all 1:1000, and signal developed with ECL prime western blotting reagent (GE Helthcare, United Kingdom). Please see Supplementary Fig. 8 for uncropped immunoblots.

**Reversible biotinylation assay.** Cells (400,000) transiently expressing DAT were seeded in six well plates 1 day prior to the experiment. On the day of the experiment, the cells were incubated for 30 min with 1.2 mg ml$^{-1}$ sulfo-NHS-S-S-biotin in PBS on ice and washed twice in ice-cold 100 mM glycine in PBS to remove unbound sulfo-NHS-S-S-biotin. During the internalization period, the samples were treated with mβCD or left untreated as described above. Two wells were kept on ice and one well was incubated for 30 min at 37 °C in cell media without FBS supplemented with 100 µg ml$^{-1}$ leupeptin and 20 µM monensin. To strip the biocytin from the surface all samples but the "total surface" sample were incubated twice in 100 mM MesNA, 0.2% bovine serum albumin, 50 mM Tris-HCl, pH 8.8, 100 mM NaCl, 1 mM EDTA and subsequently washed twice in ice-cold PBS. Samples were lyzed and mixed for 20 min at 4 °C in lysis buffer (25 mM Tris, pH 7.5, 100 mM NaCl, 1 mM EDTA, 1% Triton X-100, 0.2 mM PMSF, protease inhibitor mixture (Roche Applied Science), and 5 mM $N$-ethyl-maleimide) and afterwards centrifuged for 15 min at 16,000 × g. We collected "total sample" from each sample and mixed with the same amount of 2× loading buffer (1% SDS, 2.5% β-mercaptoethanol, and 100 mM dithiothreitol). We pulled down the proteins labeled with bocytin over night at 4 °C using avidin beads and washed the sampled four times in lysis buffer. The samples werre centrifuged for 3 min at 3000 × g per wash. Each sample was mixed with 2× loading buffer and loaded on a 10% pre-cast gel (Bio-Rad), followed by western blotting and development as described under sucrose gradient. We utilized a mouse HRP-conjugated anti-β-actin antibody (Sigma, A3854) 1:20,000. Immunoreactivity was quantified using ImageJ. Statistical significance was determined by one-way ANOVA followed by Bonferroni's multiple comparisons test.

**Internalization assay.** Cultured dopaminergic neurons were used at 11–13 DIV. DAT at the cell surface was labeled by washed the neurons once in ice-cold imaging buffer, and incubate for 20 min with 20 nM JHC1-64 at 4 °C. The neurons were subsequently washed once in 37 °C imaging buffer before incubation with control or 5 mM mβCD in imaging buffer. After 30 min the cells were washed twice and incubated for 30 min in the imaging buffer at 37 °C before imaging. The images were thresholded uniformly. We manually defined and measured the integrated signal from total soma signal as well as the signal from the internalized fraction alone for each image. We then calculated the amount of internalized DAT as the percentage of the total soma signal.

**Super-resolution setup.** For super-resolution imaging, we used an ECLIPSE Ti-E epifluoresence/TIRF microscope (NIKON, Japan) equipped with 405 nm, 488 nm 561, and 647 nm lasers (Coherent, California, USA). All lasers are individually shuttered and collected in a single fiber to the sample through a 1.49 NA, ×100, apochromat TIRF oil objective (NIKON). For PALM imaging the activation and excitation lasers light (401 and 488 nm, respectively) are reflected by a dichroic mirror (zt 405/488/561, F68-410, AHF, Germany) and the emitted fluorescence from Dronpa is filtered by a longpass (488 Longpass, F76-490, AHF) and a bandpass filter (F47-525, AHF). For STORM imaging the activation and excitation lasers (401 and 647 nm, respectively) are reflected by a dichroic mirror (z405/488/561/647 rpc) and the emitted light from Alexa647 is filtered by a 710/80 nm bandpass filter (NIKON). For dual-color STORM we used a dichroic mirror with the range 350–412, 485–490, 558–564, and 637–660 nm (97,335 QUAD C-NSTORM C156921). The excitation light was filtered at the wavelengths: 401 ± 24 nm, 488 ± 15 nm, 561 ± 15 nm, 647 ± 24 nm. The emitted light was filtered at the wavelengths: 425–475, 505–545, 578–625, and 664–787 nm, and secondly by an extra filter to decrease noise (561 nm Longpass, Edge Basic, F76-561, AHF). Images were recorded with an EM-CCD camera (iXon3 897, Andor, United Kingdom). To minimize sample drift in the z-direction over time a motorized piezo stage controlled by a near-infrared light-adjusted perfect focus system (NIKON) is applied to the system.

**PALM.** For PALM imaging of Dronpa-DAT, we obtained 5000 consecutive frames with a frame rate of 33 Hz to construct one image. The 488 nm excitation laser was held constant, 0.4 kW cm$^{-2}$, during the capture of the image while the 405 nm activation laser was gradually increased to <0.1 kW cm$^{-2}$. To assess the role of cholesterol and the cytoskeleton, CAD cells transiently expressing Dronpa-DAT were treated with mβCD or CytD as described above.

**STORM.** STORM or dSTORM imaging was performed in a buffer containing β-mercaptoehtanol and an enzymatic oxygen scavenger system (10% (w/v) glucose, 1% (v/v) beta-mercaptoethanol, 50 mM Tris-HCl (pH 8), 10 mM NaCl, 34 µg ml$^{-1}$ catalase, 28 µg ml$^{-1}$ glucose oxidase), which increases photoswitching by generating a long-lived off state for Alexa647 required for STORM. For single color STORM imaging, the high intensity 647 nm excitation laser was held constant, 2.3 kW cm$^{-2}$, and the low intensity 405 nm activation laser was gradually increased to <0.1 kW cm$^{-2}$ during the experiment. 10,000 cycles of four consecutive frames (one activation frame and three consecutive imaging frames) were acquired with a cycle rate of 56 Hz. For single color dSTORM imaging, super-resolution images were constructed from 5000 frames, where the sample was subject to constant high intensity 647 nm laser and incrementally increased low intensity 405 nm laser light. For dual-color dSTORM imaging, we acquired 10,000 cycles of one frame of 561 nm laser activation followed by one frame of 647 nm laser at 16 Hz per cycle. Shutters regulated the light path in order to separate the light between the frames. The 561 nm and 647 nm lasers were held constant at 0.6 and 1.1 kW cm$^{-2}$, respectively, while a 405 nm laser was gradually increased to <0.1 kW cm$^{-2}$ in order to gradually transfer the reporter dyes from the dark state to an active state. For quantitative single color STORM imaging we corrected for multiple detection of single events by using an acquisition scheme of four frames in one image cycle: one activation frame and three subsequent imaging frames and the built analysis in Nis-Elements (NIKON N-STORM). We filtered particles so only particles that showed up in the first imaging frame and were gone by the third imaging frame were fitted as one specific particle. Particles that appeared in other imaging frames or remained for longer periods were not fitted. Secondly, we removed any event appearing in the same pixel plus-minus one pixel in both two dimensions in the consecutive image cycle. For dSTORM imaging, we corrected for events occurring in consecutive frames using the built-in tool in NIS-Elements (NIKON). This automated analysis combines events maintained throughout several frames and combines these events as one.

**Super-resolution data analysis.** Particles were fitted using the built-in STORM module in NIS-Elements (Nikon) using a least-squares fitting methods to the raw data. Particles appearing in more than one frame was removed from consecutive frames and only counted once. For Dronpa (live-PALM) particles were only fitted when having an 8× signal-to-noise ratio and filtered by having at least 200 photons. For Alexa647 and CF568 (STORM) particles were required a 10× signal-to-noise ratio and 1000 photons. The localization precision was calculated as described[64] and the resolution was assessed using the LocAlization Microscopy Analyzer (LAMA)[65, 66]. For PALM imaging of Dronpa-DAT the localization precision was ~15 nm whereas we achieved a localization precision of ~ 7 nm for STORM imaging (Alexa647). For STORM images, LAMA indicated a resolution of ~ 65 nm. To correct for drift in the X/Y direction, we applied the built-in drift correction in the analysis tool NIS-elements (NIKON). Using continuous bright particles as fiducial markers this corrects for drift occurring in the sample due to slow temporal acquisition.

**Density-based clustering analysis of super-resolution data.** For the cluster analysis we used DBSCAN[67]. This density-based clustering algorithm searches for clusters by looking for number of localizations within a circle defined by its radius, ε, and its center, p. If the area of the circle contains more than a minimum number of points (MinPts), a new cluster with p as a core object is created. Thus, the DBSCAN algorithm requires the definition of the two input parameters ε and MinPts. The algorithm then subsequently tests all the localizations found in the first cluster for having MinPts within a radius, ε, and assigns them to the cluster, and so forth. If a localization does not have MinPts within ε, this point is defined as a border point of the cluster. We carefully determined ε and MinPts. We chose ε = 30 nm in order to achieve a search radius that would allow more than one DAT protein with antibodies bound next to each other in the plasma membrane inside the cluster. MinPts was chosen based on an evaluation of number of points in smaller clusters in our images. For our super-resolution data we applied custom routines and flexible procedures for clustering (fpc) package for R. We filtered the images by removing the monomer fraction localization with (ε = 60 and MinPts = 10). Secondly, we defined the cluster fraction (ε = 30 and MinPts = 20) and generated cluster maps and assigned single colors to the individual clusters. For Dronpa-DAT data we defined the cluster fraction of ε = 30 and MinPts = 5 (For nystatin experiments both control and nystatin treated cells were analyzed with MinPts = 3.). We tested the values on our images to determine DBSCAN's ability to assign meaningful clusters. For the statistical analysis of the data we applied either a student's t-test or a one-way ANOVA combined with Bonferroni's post-test.

Cluster size was determined using custom written scripts. The cluster area was identified by first finding the convex hull of each cluster through Andrew's monotone chain convex hull algorithm. The area was calculated from the convex hull localizations by Gauss's area formula, also known as the shoelace algorithm or the surveyor's area formula[68]. Finally, the cluster sizes were fitted using a lognormal distribution in Matlab 2015a (Matlab, Mathworks).

**Coordinate based co-localization analysis.** For the co-localization analysis of the dual-color DAT stainings, we used a coordinate based colocalization (CBC)

analysis within the super-resolution quantification tool, LAMA[65, 66]. This method of analysis utilizes the coordinate information from each localization, rather than the intensity of the signal, in order to determine how much a given biological signal is distributed in reference to a separate biological signal, and vice versa. Each localization is assigned a CBC value between −1 and 1 that is determined by the distribution of localizations arising from the same source and from the compared source. Final CBC values of 1 indicate that the localization is part of a heterogenous group from the two populations. CBC values near 0 indicate that the localization is in a homogenous group that is not near any localizations from the other population. CBC values of −1 indicate that the localization is part of a homogenous group of localizations that is in the vicinity of the other localization population. This entire process was automated and completed through the LAMA toolkit[66].

**Nanocluster verification**. The validity of the nanocluster phenotype in the PALM studies was confirmed through the use of an analysis method published by Baumgart et al.[24]. This approach requires a data set containing a large range of expression levels in order to discern true clustering vs. cluster artifacts from stochastic blinking. A cluster mask was first applied to each image, where each localization was used as the center for a Gaussian, and a threshold was applied to identify a binary cluster mask. The Gaussians had a standard deviation of 80 nm and were cut off at 40 nm from the center. These parameters were identified based off visual inspection. A threshold of 2.5 was utilized to identify clusters, as recommended in the publication of this method[24]. From each image, the relative area covered by the cluster mask ($\eta$) and the normalized average density of localizations within the clusters ($\rho/\rho_0$) were found, and compared in order to verify true clustering. For comparison, the expected values for a random distribution, as found by Baumgart et al., are given by the equation:

$$\rho/\rho_0 = 1 + 1.4\eta^4. \tag{1}$$

Data positively deviating from the line for a random distribution indicates that true nanoclusters are identified.

The validity of the nanocluster phenotype in the STORM studies was supported by a nearest-neighbor analysis. Through an in-house written python script, the localizations were organized into a KDTree with the use of the python library scipy. spatial. The number of other localizations present within a 30 nm radius from each localization was identified through this KDTree structure, and the values were indexed with the coordinates of each localization. Images utilizing nearest-neighbor analysis were constructed utilizing this number of neighbor value as the color index and/or as the z axis value in order to represent the density of the localizations within clusters.

**Statistical analyses**. Imaging experiments were in general carried out three times on multiple cells (numbers indicated for each experiments) to ensure reproducible effect sizes. Biochemical experiments were carried out at least three times to ensure reproducibility. To avoid any bias and to further increase reproducibility samples were purposely kept in random orders when performing stainings, treatments as well as microscopy. Data were analyzed using unpaired, two-tailed t-tests, one-way analysis of variance (ANOVA) or repeated measures two-way ANOVA, wherever appropriate. Bonferroni's post-hoc test was performed following significance with an ANOVA. We tested for variance in data showing that no significant difference was observed between groups compared in the statistical tests. Prism software was used for statistical analysis (GraphPad Inc, Version 6, La Jolla, CA, USA). Data are presented as means ± s.e.m. Significance level was set at $P < 0.05$.

**Code availability**. Custom written codes used for the data analysis can be found at: https://github.com/GetherLab/Super-Resolution-Data-Analysis together with a description. For the analysis we have used a combination of the two programs R and Python. All parameters are explained above.

**Data availability**. The authors declare that all the data supporting the findings of this work are available with the article and its Supplementary Information files and available from the corresponding author upon reasonable request.

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

## Acknowledgements

We thank Anette Dencker Kaas for excellent technical assistance. We thank Dr. Amy H. Newman for providing JHC 1-64. The work was supported by the National Institute of Health Grants P01 DA 12408 (UG), the UNIK Center for Synthetic Biology (UG), Lundbeck Foundation Center for Biomembranes in Nanomedicine (UG), the Danish Council for Independent Research–Medical Sciences (UG, FHH), The Faculty of Health and Medical Sciences, University of Copenhagen (Ph.D. fellowship to T.R.-C.) and the Novo Nordisk Foundation (UG).

## Author contributions

T.R.-C. and M.D.L. conducted all the experiments with help from M.A. T.R.-C., J.E., F.V., T.N.J., F.H.H. and U.G. designed the experiments. T.R.-C., M.D.L., S.E. and U.G. analyzed the data. T.R.-C., M.D.L., S.E., F.H.H. and U.G. wrote the manuscript.

## Additional information

**Competing interests:** The authors declare no competing financial interests.

