## [Peer review file · Nature Communications]

Reviewers' comments:

Reviewer #1 (Remarks to the Author):

The study by Rahbek-Clemmensen and collaborators explores nanoscale membrane distribution of dopamine transporter (DAT) in the Cath.a-differentiated cells (CAD cell) and in the cultured dopamine neurons. The existence and the nature of nanoscale membrane distribution of dopamine transporter have remained enigmatic, because, these membrane domains are too small for visualization by conventional fluorescence microscopy. The group employed STORM and PALM super-resolution microscopy to investigate the localization of endogenous DAT in the membrane of dopamine neurons. The impressive STORM images have visualized the DAT distribution in the soma, neuronal extensions and presynaptic varicosities. The data strongly support the idea that the DAT molecules are localized to discrete, irregular, cholesterol-dependent nanodomains with diameters of around 200 nm. STORM and PALM experiments showed KCl-induced (40 mM - for 30 min) depolarization decreased nanodomain distribution of DAT in both dopamine neurons and in the CAD cells; whereas, [redacted].

This study could be improved by including the followings:

1. The authors should further explore the effect of increased membrane cholesterol (Cholesterol load condition) on the size and distribution of DAT in these nanodomains.
2. The discussion should confer why disruption of the actin polymerization with cytochalasin D did not affect nanoscale membrane distribution of DAT. This is specifically important since actin organization (polymerization/depolymerization) and membrane depolarization are often going hand in hand.

Overall, this is a novel study; it is very well done and is interesting. The authors elegantly took advantages of multiple techniques to study nanoscale distribution of DAT at the membrane of dopamine neurons.

Reviewer #2 (Remarks to the Author):

In this manuscript by Rahbek-Clemmensen and colleagues, the authors took advantage of super-resolution microscopy techniques (STORM and PALM) and related image analysis techniques to examine the distribution of the membrane dopamine transporter (DAT) in the CAD cell line or in rat dopamine neurons in culture. The authors' main conclusion is that the DAT tends to form membrane clusters with a diameter of approximately 200 nm and that treating the cells with methyl-beta-cyclodextrin to deplete membrane cholesterol, or long-term depolarization of membranes using high potassium saline leads to a decrease in signal intensity without a change in cluster size.

This manuscript presents very interesting novel results from experiments that appear to have been expertly carried out. However, the breath of the scientific conclusions that can be reached from these results is not so clear to me. First of all, the introduction of the manuscript does not provide a clear and compelling rationale for the experiments. What was the underlying biological question that the authors wanted to tackle? And why focus specifically on the contribution of cholesterol or on the impact of chronic membrane depolarization? What hypothesis did the authors want to test? The manuscript would gain by providing a stronger rationale in the introduction (and abstract). Also, what the major insights that derive from these data are is not so clear to me. How does improved knowledge on the size of clusters in which DAT is found on the membrane clarify current knowledge or guide future research? And is this information on cluster size special for DAT and related proteins or is this simply reflective of the typical size of non-synaptic membrane rafts in

neurons?

A few other issues to consider :

1. The conclusions of previous electron microscopy data on the localization of DAT in neurons should be more clearly discussed in the introduction and/or discussion. What was unclear based on previous data that needed to be clarified by approaches such as super-resolution microscopy?
2. The authors conclude that the nanoscale distribution of DAT may serve a key role in controlling DAT activity and availability in dopamine neurons. The reason why the authors believe this should be explained.
3. Do the CAD cells used show a negative membrane potential like neurons? And does 40 mM potassium lead to a major membrane depolarization in these cells? I am not sure why the authors chose to examine the impact of membrane depolarisation in such a cell line.
4. The protocol used to evaluate the impact of membrane depolarization on DAT distribution is unfortunately completely non-physiological. I am not sure what one can conclude from studying the impact of a 30 min, constant membrane depolarization. The authors should probably try to use electrical field stimulation or optogenetic train stimulation.
5. The authors contend on page 8 that the decrease in cluster intensity induced by methyl-beta-cyclodextrin or by long-lasting depolarisation is not due to enhanced transporter internalization. It would be more convincing to include a positive control demonstrating that the biotinylation assay used is effective to detect internalization of DAT induced by other signals previously known to induce internalization.
6. The example shown of a membrane invagination containing DAT signal and that could represent an endocytosis or exocytosis event (Fig. 4g, h) is purely anecdotal and should be removed from the manuscript or further studied, quantified and manipulated.
7. The experiment performed with TH (Supplementary figure 5) is an important control and I suggest that it be further documented and included in the paper. The cluster map and quantitative analysis should be presented. If the size of the clusters turns out to be similar to that of DAT, what would the authors conclude?
8. The JHC-1-64 data presented in supplementary figure 7 is quite interesting. It shows very clear membrane DAT labelling at the cell body level. This contrasts with the STORM data, that the authors interpret as revealing very low levels of somatic DAT. This is puzzling and somewhat contradictory. It should be discussed.
9. There are problems with the lettering of some of the axis labels of some of the figures (strange symbols that seem to have appeared in the PDF version of the figures).

Reviewer #3 (Remarks to the Author):

The manuscript by Gether and coworkers study the organization of the dopamine transporter at the plasma membrane using single-molecule super-resolution microscopy and quantitative data analysis.

First of all, many graphs are incorrectly labeled - there seems to be a font problem in the .pdf. This holds for Figures 1, 2, 5 and 6, and I am not able to interpret this data.

On the biological side, my main concern is that applying MCD is interpreted as cholesterol depletion, without sufficient control. MCD has a number of drastic effects on a cell, and any response of a cell (or in this case the reorganization of a receptor) cannot simply be attributed to a single effect. The authors write: "we depleted cholesterol [...] using MCD", that is wrong. The author added MCD to their cells, which - among many other effects - reduces the content of cholesterol in the plasma membrane. This clearly requires more experimental controls and also other drugs, nystatin might be an option. This comment is equally applicable to cytochalasin.

On the technological side, I have major concerns with the data analysis and interpretation. That starts with the fact that I could not find any information on the experimental localization precision, or drift correction (x/y). The cluster analyses have even more serious issues:

- Ripley analysis (Fig 1, 2) reports large clusters that are definitely not found in the PALM images

shown; with the scale bar being 500 nm, I cannot find clusters of 200 nm or more.

- It appears to me as if the authors interpreted H_{\max} as the cluster diameter. This is wrong, see e.g. Malkusch et al., Histochem Cell Biol 2013, PMID 22910843.

- in Fig 3, the authors switch to DBSCAN analysis - why not analyzing all data with DBSCAN right away? Multiple events of photoswitching (as in dSTORM) can be corrected for in DBSCAN. (NB this should also be done for the Ripley analysis by using a spatio-temporal grouping filter, which I could not find any detail in this manuscript - the authors only mention that they did, without any details)

- what is the rationale for choosing the value for the epsilon parameter for DBSCAN?

- in Figure 4m/4o, a color code highlights clusters. If these are the clusters determined, then the DBSCAN parameters were set incorrectly. Clearly, within the "red clouds", there are multiple clusters which are assigned as one. It is helpful to color-assign clusters.

- DBSCAN is described insufficiently: it is not clear

- the authors used the dSTORM method throughout, and should name it this way

- methods, the software likely determined the "localization precision" (and not the accuracy), and for sure did not determine the resolution

- methods, Ripley analysis: there is no "estimation" in the Ripley function, it simply counts the number of particles in concentric areas around each localization (page 24). The authors also did not use "derivatives of Ripley's K function" (a derivative is a mathematical operation), they used variants of it (H and L function).

- methods, DBSCAN: what is the rationale for choosing the epsilon parameter of 60? Epsilon has a unit (distance), did the authors mean "60 nm"? If so, the value is quite high and might explain that the analysis returns larger cluster than those that can be seen in the images.

I think it is very critical to extract numbers from super-resolution data this way, without a proper discussion on why parameters were chosen the way they are. In particular, data analysis should match what we see in the images.

Page 5, the statement of "~20 nm resolution" is both wrong and lacks experimental data. The authors might have determined the localization precision, and if so, the authors should say how (e.g. Rieger formula, or experimental by nearest neighbor analysis etc.) and this value should be given here. The "resolution" depends on many other factors as well, e.g. labeling density (Nyquist).

Further comments

- Page 6, "Thus, as both ...", incomplete sentence

- The description of DBSCAN on page 9 is wrong. DBSCAN has a distance parameter, epsilon, and a N_{\min} parameter, and by that classifies each localization into a core, a neighbor or an outside localization. There is no "distance to nearest neighbor" parameter.

- Name fluorophores consistently throughout manuscript

- Methods, the authors should specify whether "10% Glucose" refers to (w/w), (w/v) etc.

- Methods/STORM, this is not a "mercaptoethanol buffer"; also, this buffer does not "enhance blinking", it generates the long-lived off-state that is required

- Methods/STORM, irradiation densities for both lasers are missing

RESPONSE TO REVIEWERS (NCOMMS-16-00358)

We acknowledge the very careful evaluation of our manuscript and the many constructive comments from the reviewers. As outlined below we have now dealt with all issues and criticisms as well as we have included a substantial amount of new data.

Reviewer 1

“The study by Rahbek-Clemmensen and collaborators explores nanoscale membrane distribution of dopamine transporter (DAT) in the Cath.a-differentiated cells (CAD cell) and in the cultured dopamine neurons. The existence and the nature of nanoscale membrane distribution of dopamine transporter have remained enigmatic, because, these membrane domains are too small for visualization by conventional fluorescence microscopy. The group employed STORM and PALM super-resolution microscopy to investigate the localization of endogenous DAT in the membrane of dopamine neurons. The impressive STORM images have visualized the DAT distribution in the soma, neuronal extensions and presynaptic varicosities. The data strongly support the idea that the DAT molecules are localized to discrete, irregular, cholesterol-dependent nanodomains with diameters of around 200 nm. STORM and PALM experiments showed KCl-induced (40 mM - for 30 min) depolarization decreased nanodomain distribution of DAT in both dopamine neurons and in the CAD cells; [redacted].

This study could be improved by including the followings:

1. The authors should further explore the effect of increased membrane cholesterol (Cholesterol load condition) on the size and distribution of DAT in these nanodomains.”

We appreciate this constructive suggestion; however, to further investigate the role of cholesterol we decided to follow the suggestion by Reviewer 3 and use nystatin, a cholesterol-binding polyene antibiotic that sequester cholesterol, to assess the importance of cholesterol through a different mechanism compared to methyl- β -cyclodextrin (ref 27). Indeed, nystatin decreased nanodomain localization of DAT in both transfected CAD cells and in dopaminergic neurons. *The new data are for the CAD cells shown in Supplementary Figure 3 and described on page 6, 1st paragraph. For the neurons, the data are shown in Supplementary Fig. 5 and the data described on page 9, bottom.*

“2. The discussion should confer why disruption of the actin polymerization with cytochalasin D did not affect nanoscale membrane distribution of DAT. This is specifically important since actin organization (polymerization/depolymerization) and membrane depolarization are often going hand in hand.”

We are a little confused by this comment. Since cytochalasin D had no effect, we find it reasonable to conclude that the nanodomain distribution of DAT does not depend on the cytoskeleton. Although we recognize that actin polymerization can be linked to membrane depolarization, we do not find that the lack of such a connection particularly surprising, and in the absence of an effect of cytochalasin D, it was not investigated further.

“Overall, this is a novel study; it is very well done and is interesting. The authors elegantly took advantages of multiple techniques to study nanoscale distribution of DAT at the membrane of dopamine neurons.”

We strongly appreciate these very positive comments from this reviewer.

Reviewer 2

We also strongly appreciate the overall positive comments from this reviewer, as well as we acknowledge that the biological scope may not have been entirely clear in the first version of the manuscript. As outlined below we are convinced that this criticism has been addressed in the new version.

“In this manuscript by Rahbek-Clemmensen and colleagues, the authors took advantage of super-resolution microscopy techniques (STROM and PALM) and related image analysis techniques to examine the distribution of the membrane dopamine transporter (DAT) in the CAD cell line or in rat dopamine neurons in culture. The authors' main conclusion is that the DAT tends to form membrane clusters with a diameter of approximately 200 nm and that treating the cells with methyl-beta-cyclodextrin to deplete membrane cholesterol, or long-term depolarization of membranes using high potassium saline leads to a decrease in signal intensity without a change in cluster size.

This manuscript presents very interesting novel results from experiments that appear to have been expertly carried out.

However, the breadth of the scientific conclusions that can be reached from these results is not so clear to me. First of all, the introduction of the manuscript does not provide a clear and compelling rationale for the experiments. What was the underlying biological question that the authors wanted to tackle?

And why focus specifically on the contribution of cholesterol or on the impact of chronic membrane depolarization? What hypothesis did the authors want to test? The manuscript would gain by providing a stronger rationale in the introduction (and abstract). Also, what the major insights that derive from these data are is not so clear to me. How does improved knowledge on the size of clusters in which DAT is found on the membrane clarify current knowledge or guide future research? And is this information on cluster size special for DAT and related proteins or is this simply reflective of the typical size of non-synaptic membrane rafts in neurons?”

These are all important and relevant questions. In the new version of the manuscript, we have addressed the questions and attempted to better explain the rationale for our study (*please see revised Abstract, Introduction and Discussion*). The physiological hypothesis underlying our study is that DAT is subject to temporal and spatial regulation that is not readily detectable by classical means. Thus, classical “bulk measurements” may not be able to detect nanoscale heterogeneities in e.g. subcellular distribution that could play a hitherto

unknown key role in spatiotemporal control of transporter function. Such a hypothesis warrants investigation with novel techniques such as super-resolution microscopy. Importantly, our data suggest that the hypothesis is correct. That is, we find evidence that DAT is dynamically distributed into cholesterol-sensitive nanoscale domains. The existence of such small domains (that notably are invisible by traditional microscopic technique) might enable the neuron to rapidly switch the transporter between different functional localizations and thereby optimize availability and activity of the transporter in the presynaptic terminals.

This becomes even more interesting given our new finding that brief stimulation of ionotropic NMDA-type glutamate receptors reversibly decreased nanodomain localization of DAT (*data are shown in a new Figure 5 and the data are described on page 10*). The physiological basis for mimicking the effect of excitatory input was simply that enhanced neuronal activity conceivably could lead to altered spatial demands for reuptake capacity and thus possibly would be a yet poorly understood regulator of DAT function. It should also be mentioned that in a recent elegant paper by Khoshbouei and co-workers it was suggested that membrane potential can regulate surface trafficking of DAT (ref. 9).

As for the Reviewer's question about our focus on cholesterol, we find that the physiological rationale for assessing cholesterol-dependence is rather strong. First, previous reports have suggested that DAT is distributed to cholesterol- and glycosphingolipid-enriched plasma membrane micro domains ("membrane rafts") and it has been proposed, based on biochemical experiments, that localization to such rafts in the plasma membrane can regulate DAT trafficking as well as amphetamine-induced efflux (ref 12 and 13). Second, the crystal structure of the drosophila DAT revealed a cholesterol-binding site in the transporter (ref 38), suggesting direct regulation of the transporter by cholesterol. We therefore hypothesized that the nanodomains identified by super-resolution microscopy displayed sensitivity to cholesterol. *The importance of cholesterol is described on page 3, bottom and page 4, top.*

To assess the uniqueness of the nanoscale distribution of DAT, we have in the new manuscript included a new series of dual-color dSTORM experiment. These new data reveal important insight into the molecular architecture of the putative dopaminergic presynapse by describing the nanoscale distribution of DAT relative to two other key components of dopaminergic terminals, tyrosine hydroxylase (TH) and vesicular monoamine transporter 2 (VMAT2) (*data are shown in two new Figures 6 and 7, and the data are described on pages 11-12*). The data demonstrate a distinct distribution of these proteins in the presynaptic terminals, and reveal how VMAT2, as well as TH, are localized immediately adjacent to, but not overlapping with, the cholesterol-enriched DAT nanodomains (*see also Discussion on page 15-16*).

Summarized, we find that our manuscript indeed conveys important biological information and also guides future research. Future important research goals will not least include further dissection of the molecular architecture of the dopaminergic presynapse to understand in detail how the nanoscale distribution of DAT and other key proteins (including DAT associated proteins) adapts to different functional states, how it contributes

to dopamine homeostasis and how it might change during disease. Please also see the last paragraph of the Discussion on page 17.

“A few other issues to consider :

1. The conclusions of previous electron microscopy data on the localization of DAT in neurons should be more clearly discussed in the introduction and/or discussion. What was unclear based on previous data that needed to be clarified by approaches such as super-resolution microscopy?”

The conclusions of previous EM studies are now more clearly stated in the Introduction (see page 3, 2nd paragraph). The EM studies elegantly described the general localization of DAT in dopaminergic neurons. However, although EM offers superior resolution, technical difficulties and requirements for sample preparation challenge the applicability of EM when testing the effect of pharmacological and biochemical manipulations, as done in the present study. Specifically, low labeling efficiency of immunogold EM imaging makes the technique less suitable for investigating and quantifying protein clustering. The issue is discussed on page 13, 1st paragraph of the Discussion.

“2. The authors conclude that the nanoscale distribution of DAT may serve a key role in controlling DAT activity and availability in dopamine neurons. The reason why the authors believe this should be explained.”

Please read our response above to the general issues raised by the Reviewer.

“3. Do the CAD cells used show a negative membrane potential like neurons? And does 40 mM potassium lead to a major membrane depolarization in these cells? I am not sure why the authors chose to examine the impact of membrane depolarisation in such a cell line.”

We acknowledge this criticism and have now taken the potassium data out of the manuscript.

“4. The protocol used to evaluate the impact of membrane depolarization on DAT distribution is unfortunately completely non-physiological. I am not sure what one can conclude from studying the impact of a 30 min, constant membrane depolarization. The authors should probably try to use electrical field stimulation or optogenetic train stimulation.”

We agree that depolarization by use of potassium is not truly physiological (although the treatment is often used) and have accordingly removed the potassium data from the manuscript.

“5. The authors contend on page 8 that the decrease in cluster intensity induced by methyl-beta-cyclodextrin or by long-lasting depolarisation is not due to enhanced transporter internalization. It would be more convincing to include a positive control demonstrating that the biotinylation assay used is effective to detect internalization of DAT induced by other signals previously known to induce internalization.”

The biotinylation assay is well established and used many times by our lab as well as other labs. We strongly believe based on previous experience that increased internalization – if it had occurred - would have been revealed by the procedure employed.

“6. The example shown of a membrane invagination containing DAT signal and that could represent an endocytosis or exocytosis event (Fig. 4g, h) is purely anecdotal and should be removed from the manuscript or further studied, quantified and manipulated.”

As generally accepted in imaging studies, we describe in detail what is observed in the images, but we do not quantify and manipulate each single observation. We allowed ourselves to show an example of a membrane invagination that may represent an endocytosis event (Figure 3m), as well as we show a structure in the same figure that could be a DAT containing vesicle (Figure 3h). Importantly, it would not be possible to visualize these structures with such detail with classical fluorescence microscopy. At the same time, we do acknowledge that we neither sought to quantify nor manipulate the structures. However, we still find that they deserve to be shown as they indicate the potential strength of the super-resolution technique. It is also important to stress that no major conclusion of the paper is based on the two shown structures.

“7. The experiment performed with TH (Supplementary figure 5) is an important control and I suggest that it be further documented and included in the paper. The cluster map and quantitative analysis should be presented. If the size of the clusters turns out to be similar to that of DAT, what would the authors conclude?”

We agree that visualizing other related proteins in the dopaminergic neurons is not only an important control but also scientifically very interesting. As described in detail above, the manuscript now includes detailed dual dSTORM imaging comparing the distribution of DAT with that of both TH and VMAT2.

“8. The JHC-1-64 data presented in supplementary figure 7 is quite interesting. It shows very clear membrane DAT labelling at the cell body level. This contrasts with the STORM data, that the authors interpret as revealing very low levels of somatic DAT. This is puzzling and somewhat contradictory. It should be discussed.”

When labeling endogenous DAT in cultured neurons or brains slices using immunocytochemical or immunohistochemical approaches, we have observed a high degree of DAT in the intracellular compartments of the somas, relative to the signal observed from the

plasma membrane. However, when labeling DAT with JHC 1-64, we only see surface DAT as it binds to the transporter without passing the plasma membrane (unless an internalization assay is performed) (refs. 8, 10, 31). The two approaches therefore look at two different pools of DAT. As a consequence, the amount of DAT present on the cell surface might be less apparent in the STORM images compared to the JHC 1-64 confocal images. We should also note that when comparing JHC1-64 images of soma versus extensions we do see generally higher plasma membrane signal from the extensions than from the somas although this is not clear from the images shown in Supplementary Figure 6. *The issue is now discussed on page 10, 1st paragraph.*

“9. There are problems with the lettering of some of the axis labels of some of the figures (strange symbols that seem to have appeared in the PDF version of the figures).”

We apologize for these mistakes. They have all been corrected in the new manuscript.

Reviewer 3

We acknowledge the thorough technical review of our data and have now dealt with all comments as outlined below.

“The manuscript by Gether and coworkers study the organization of the dopamine transporter at the plasma membrane using single-molecule super-resolution microscopy and quantitative data analysis. First of all, many graphs are incorrectly labeled - there seems to be a font problem in the .pdf. This holds for Figures 1, 2, 5 and 6, and I am not able to interpret this data.”

We sincerely apologize for this inconvenience. It happened during file conversion for the upload. All labeling has been checked and corrected in the new manuscript.

“On the biological side, my main concern is that applying MCD is interpreted as cholesterol depletion, without sufficient control. MCD has a number of drastic effects on a cell, and any response of a cell (or in this case the reorganization of a receptor) cannot simply be attributed to a single effect. The authors write: "we depleted cholesterol [...] using MCD", that is wrong. The author added MCD to their cells, which - among many other effects - reduces the content of cholesterol in the plasma membrane. This clearly requires more experimental controls and also other drugs, nystatin might be an option. This comment is equally applicable to cytochalasin.”

This is an important comment and we agree that depletion of cholesterol with methyl- β -cyclodextrin (m β CD) can have strong effects (e.g. cause cell death), although it is a widely used method and is a generally accepted way of depleting cholesterol from the plasma membrane (see e.g. ref 26). To address the concern, we have now assessed the effect of

nystatin, a cholesterol-binding polyene antibiotic, which can sequester cholesterol in the membrane and disrupt membrane rafts (see e.g. ref 27). Nystatin treatment reduced DAT nanodomain localization both in Dronpa-DAT expressing CAD cells (assessed by PALM) and in dopaminergic neurons (assessed by STORM). It is also important to note that we observed no morphological changes or signs of cell death in response to m β CD during the time frame of our experiments. The new results for the CAD cells shown in a new Supplementary Figure 3 and described on page 6, 1st paragraph. For neurons, the results are shown in a new Supplementary Figure 5 on page 9, bottom.

“On the technological side, I have major concerns with the data analysis and interpretation. That starts with the fact that I could not find any information on the experimental localization precision, or drift correction (x/y).”

We have calculated the localization precision for both the PALM and the STORM/dSTORM data according to Thompson et al 2002 (ref 63). The experimental localization precision is 7 nm for our PALM data and 15 nm for our STORM/dSTORM data. This is now described in the Methods section of the manuscript, section ‘Super-resolution data analysis’ on page 24.

To correct for drift during the experiments, we applied the build-in drift correction in the NIKON N-STORM system to the data analysis. The module analyzes any stable events, present in several frames, over time and uses them as fiducial markers. This allows for drift correction of the image without adding fiducial markers such as gold nanoparticles or TetraSpeck Microspheres. This is now described in the Methods section of the manuscript, section ‘Super-resolution data analysis’ on pages 24-25.

“The cluster analyses have even more serious issues:

- Ripley analysis (Fig 1, 2) reports large clusters that are definitely not found in the PALM images shown; with the scale bar being 500 nm, I cannot find clusters of 200 nm or more.
- It appears to me as if the authors interpreted H_max as the cluster diameter. This is wrong, see e.g. Malkusch et al., Histochem Cell Biol 2013, PMID 22910843.
- in Fig 3, the authors switch to DBSCAN analysis - why not analyzing all data with DBSCAN right away?”

As suggested by the reviewer we have now removed the Ripley’s K analysis from the manuscript and replaced it throughout with the DBSCAN analysis, please see Figures 1-5, Supplementary figure 3, 5 and Methods section ‘Density-based clustering analysis of super-resolution data’ page 25.

To further strengthen our cluster analysis of the PALM data, we have implemented a recently published analysis by Baumgart et al. (ref 24) to verify that observed DAT clusters represent true biological structures. Indeed, due to the increase in localization density within detected clusters as the total clustered area increases, the analysis supported the claim that the identified clusters are not the result of multiple observations of single fluorophores. Note that this analysis could only be utilized on the PALM data as it requires the range of label density achieved through the varied expression of the fluorescent protein

construct. The analysis is show in Supplementary Figure 1 and the data described on page 5, bottom of 1st paragraph. The method is described in the Method section “Nanocluster verification” on pages 26-27.

For the STORM data we applied a nearest-neighbor density analysis, as true clusters can should be distinguishable from imaging artifacts based on the density of localization within individual clusters (ref 30). This separate method was utilized because the STORM fluorophores produce enough localizations so the differences in density can be clearly identified. Importantly, the individual clusters did not have uniform density but rather became denser at the center, supporting a truly clustered distribution. The analysis is shown in Figure 3s-v and described on page 9, 2nd paragraph. The method is described in the Method section “Nanocluster verification” on page 27.

“Multiple events of photoswitching (as in dSTORM) can be corrected for in DBSCAN. (NB this should also be done for the Ripley analysis by using a spatio-temporal grouping filter, which I could not find any detail in this manuscript - the authors only mention that they did, without any details).”

For quantitative single-color STORM imaging we corrected for multiple detection of single events by using an acquisition scheme of 4 frames in one image cycle: 1 activation frame and 3 subsequent imaging frames and the built analysis in Nis-Elements (NIKON N-STORM) We filtered particles so only particles that showed up in the first imaging frame and were gone by the third imaging frame were fitted as 1 specific particle. Particles that appeared in other imaging frames or remained for longer periods were not fitted. Secondly, we removed any event appearing in the same pixel plus-minus one pixel in both 2 dimensions in the consecutive image cycle. For dSTORM imaging, we corrected for events occurring in consecutive frames using the built-in tool in NIS-Elements (NIKON). This automated analysis combines events maintained throughout several frames and combines these events as one. This is now described in Methods section ‘STORM’ on page 24.

“- what is the rationale for choosing the value for the epsilon parameter for DBSCAN?”

We understand the concern of the Reviewer and have evaluated the ϵ parameter as well as the MinPts parameter. We have chosen ϵ to define a small diameter (30 nm) that still is biological meaningful by allowing for multiple DAT proteins with antibodies bound in the circle defined by the core, p, and the radius, ϵ . This is now explained in the Methods section ‘Density-based clustering analysis of super-resolution data’ on page 25.

“- in Figure 4m/4o, a color code highlights clusters. If these are the clusters determined, then the DBSCAN parameters were set incorrectly. Clearly, within the "red clouds", there are multiple clusters which are assigned as one. It is helpful to color-assign clusters.”

We have taken this into consideration and our cluster maps are now based on color assignment of the clusters instead of a heat map, and we hope the reviewer will find this more useful for visual representation of the clusters in Figures 1-5 and Supplementary Figure 3.

“- DBSCAN is described insufficiently: it is not clear”

Based on Reviewer’s comment we have revised the description of our DBSCAN analysis, and we hope that the Reviewer will find that the description now is clearer. Please see Methods section ‘Density-based clustering analysis of super-resolution data’ on page 25.

“- the authors used the dSTORM method throughout, and should name it this way”

Most of our single-color STORM imaging experiments rely on antibodies labeled with two fluorophores, one activator and one reporter. This procedure corresponds to the method used e.g. by Dr. Xiaowei Zhuang and co-workers who just called it “STORM” (see e.g. ref 60, and Huang et al. Science, 319, 810-813, 2008). According to Heilemann et al (Heilemann et al. Angew. Chem., 47, 6172-6, 2008), the dSTORM method involves the use of only a single fluorophore coupled to the antibody. We have used dSTORM in the following experiments: 1) the NMDA treatment experiments shown in Figure 5q; 2) the nystatin experiment shown in Supplementary Figure 5); 3) the dual-color experiments shown in Figure 6 and 7. For these experiments, we have made sure to name the technique as dSTORM.

“- methods, the software likely determined the "localization precision" (and not the accuracy), and for sure did not determine the resolution.”

We apologize for the loose use of terminology. This has now been corrected to localization precision.

“- methods, Ripley analysis: there is no "estimation" in the Ripley function, it simply counts the number of particles in concentric areas around each localization (page 24). The authors also did not use "derivatives of Ripley's K function" (a derivative is a mathematical operation), they used variants of it (H and L function).”

We agree with the reviewers on the wrongly chosen words in the description of the Ripley analysis. Moreover, Ripley’s K has now been completely replaced by DBSCAN in the manuscript.

“- methods, DBSCAN: what is the rationale for choosing the epsilon parameter of 60? Epsilon has a unit (distance), did the authors mean "60 nm"? If so, the value is quite high and might explain that the analysis returns larger cluster than those that can be seen in the images. I think it is very critical to extract numbers from super-resolution data this

way, without a proper discussion on why parameters were chosen the way they are. In particular, data analysis should match what we see in the images.”

Based on the Reviewer’s concerns, we have reevaluated our original chosen parameters for the DBSCAN analysis of $\epsilon = 60$ and MinPts = 60. Because the Reviewer is concerned that we saw multiple clusters in single clusters, we lowered the ϵ to 30 nm. At the same time we adjusted MinPts to better reflect the clustered localizations seen in the images. Moreover, the cluster map visualization has been changed throughout the manuscript. The clusters are now color-assigned, which should make it easier to visually separate the individual clusters. Importantly, the changed parameters did not change our key finding, i.e. we see a clear effect of both m β CD and nystatin treatment with the new, improved settings. For methodological details, please see Methods section ‘Density-based clustering analysis of super-resolution data’ on page 25.

“Page 5, the statement of “~20 nm resolution” is both wrong and lacks experimental data. The authors might have determined the localization precision, and if so, the authors should say how (e.g. Rieger formula, or experimental by nearest neighbor analysis etc.) and this value should be given here. The “resolution” depends on many other factors as well, e.g. labeling density (Nyquist).”

We apologize for the wrong use of the terminology. As described above, we have calculated the localization precision for both the PALM and the STORM/dSTORM data according to Thompson et al 2002 (ref 63 in manuscript). To assess resolution, we used the Localization Microscopy Analyzer (LAMA) tool, which has been described in Malkusch et al. and Malkusch & Heilemann (ref 64, 65). Using this method we find a spatial resolution of 65 nm. This is described in the Methods section ‘Super-resolution data analysis’ on page 24.

“Further comments

- Page 6, “Thus, as both ...”, incomplete sentence.”

We apologize for this. This has now been corrected in the manuscript.

“- The description of DBSCAN on page 9 is wrong. DBSCAN has a distance parameter, epsilon, and a N_min parameter, and by that classifies each localization into a core, a neighbor or an outside localization. There is no “distance to nearest neighbor” parameter.”

As stated above, we have revised the description of the DBSCAN analysis, and we hope that the Reviewer will find that the description now is clearer.

“- Name fluorophores consistently throughout manuscript.”

We apologize for this. This has now been corrected in the new manuscript.

“- Methods, the authors should specify whether "10% Glucose" refers to (w/w), (w/v) etc.”

This has been corrected.

“- Methods/STORM, this is not an "mercaptoethanol buffer"; also, this buffer does not "enhance blinking", it generates the long-lived off-state that is required.”

We agree with the Reviewer. This has now been corrected.

“- Methods/STORM, irradiation densities for both lasers are missing.”

Irradiation densities are now stated in the Method section “PALM” and “STORM” on page 23.

REVIEWERS' COMMENTS:

Reviewer #1 (Remarks to the Author):

This is a significantly revised manuscript by Rahbek-Clemmensen. The authors have commendably addressed the concerns of this reviewer. The amended analyses and additional experiments provided in the revised manuscript have strengthened the study.

Multiple complementary approaches have been expertly carried out to present very interesting and novel idea of activity dependent molecular organization of dopamine transporter in nanodomains at the membrane of dopamine neurons and heterologous expression systems.

Appropriate statistical analyses have been used.

The use of complementary molecular, biochemical and microscopy approaches in this study increases the confidence for the reproducibility of this work. Overall, this work has been expertly performed; highly sensitive approaches have been utilized to investigate an innovative concept. The reported findings will significantly advance the field of transporter biology in general and dopamine transporter in specific.

A few minor editorials points can be corrected through editorial revision (e.g. line 103 "fro", line 196 extra "in", line 496 "sentence is incomplete").

Reviewer #2 (Remarks to the Author):

The revised manuscript is greatly improved relative to the initial version.

As I previously mentioned, the illustration of a putative site of endocytosis in figure 3L and 3M is purely anecdotal and should be removed from the manuscript.

In the experiments performed with primary rat neurons, the authors refer to structures that are presumed to be axonal varicosities. When looking at images such as those shown in Fig. 4C, 4G, 4K, it appears from the scale bars that the varicosities might be something like 1 micrometer wide by 3 or 4 micrometer long (according to the scale bar). This seems awfully big for a dopaminergic axonal varicosity. The authors should comment on this and perhaps mention that the identity of such structures as axonal varicosities is speculative in the absence of co-labelling for a specific presynaptic marker (VMAT2 is not, because it is also found in the somatic and dendritic compartment).

Figure 6o does not have a scale bar.

Reviewer #3 (Remarks to the Author):

The authors addressed all my comments in a very thorough way. This has been a hard piece of work, which I do appreciate.

The manuscript has substantially improved, and I would like to see it published in Nature Communications.

RESPONSE TO REVIEWERS (NCOMMS-16-00358A-Z)

We acknowledge the very careful evaluation of our manuscript and are glad that the reviewers find our revisions satisfactory. As outlined below we have now addressed the final comments in the revised manuscript.

Reviewer 1

“This is a significantly revised manuscript by Rahbek-Clemmensen. The authors have commendably addressed the concerns of this reviewer. The amended analyses and additional experiments provided in the revised manuscript have strengthened the study. Multiple complementary approaches have been expertly carried out to present very interesting and novel idea of activity dependent molecular organization of dopamine transporter in nanodomains at the membrane of dopamine neurons and heterologous expression systems.

Appropriate statistical analyses have been used.

The use of complementary molecular, biochemical and microscopy approaches in this study increases the confidence for the reproducibility of this work. Overall, this work has been expertly performed; highly sensitive approaches have been utilized to investigate an innovative concept. The reported findings will significantly advance the field of transporter biology in general and dopamine transporter in specific.

A few minor editorials points can be corrected through editorial revision (e.g. line 103 “fro”, line 196 extra “in”, line 496 “sentence is incomplete”).”

We appreciate the positive comments from the reviewer. The editorial points have been corrected.

Reviewer 2

“The revised manuscript is greatly improved relative to the initial version.

As I previously mentioned, the illustration of a putative site of endocytosis in figure 3L and 3M is purely anecdotal and should be removed from the manuscript.

In the experiments performed with primary rat neurons, the authors refer to structures that are presumed to be axonal varicosities. When looking at images such as those shown in Fig. 4C, 4G, 4K, it appears from the scale bars that the varicosities might be something like 1 micrometer wide by 3 or 4 micrometer long (according to the scale bar). This seems awfully big for a dopaminergic axonal varicosity. The authors should comment on this and perhaps mention that the identity of such structures as axonal varicosities is speculative in the absence of co-labelling for a specific presynaptic marker (VMAT2 is not, because it is

also found in the somatic and dendritic compartment).

Figure 6o does not have a scale bar.”

We also highly appreciate the positive comments from this reviewer.

The illustration of a putative site of endocytosis in Fig. 3 has been taken out.

As for the axonal varicosities, we do not find them awfully big. In fact, we would argue that they are in the same range as should be expected. Previous confocal images from our lab roughly indicate a size of a few micrometer in cultured neurons (ref 8 in paper). Additionally, previous EM on striatal slices indicate that a presynaptic varicosity can be at least 1-2 micrometer (ref. 5). In our view, the strong enrichment of VMAT2 immunoreactivity corresponding to the varicosities in the cultured neurons also strongly support that the varicosities indeed are equivalent of presynaptic release sites in the brain - although we do admit that VMAT2 as such is not a strictly specific presynaptic marker.

We would argue that we already are careful when introducing the varicosities on page 8,2nd paragraph. Here, we write that the varicosities are “believed to equivalent to presynaptic transmitter release sites”. We have now expanded this sentence and write that the varicosities are “believed to equivalent to presynaptic transmitter release sites given the enriched presence of VMAT2” (ref 8).

A scale bar has been added to Fig. 6o.

Reviewer 3

“The authors addressed all my comments in a very thorough way. This has been a hard piece of work, which I do appreciate.

The manuscript has substantially improved, and I would like to see it published in Nature Communications.”

We are glad that the reviewer appreciates the revisions and the work we have done.